# Drying kinetics and thermo-environmental analysis of a PV-operated tracking indirect solar dryer for tomato slices

**Abdallah Elshawadfy Elwakeel**[1]*, **Mohsen A. Gameh**[2], **Awad Ali Tayoush Oraiath**[3], **I. M. Elzein**[4], **Ahmed S. Eissa**[5], **Mohamed Metwally Mahmoud**[6], **Daniel Eutyche Mbadjoun Wapet**[7]*, **Mahmoud M. Hussein**[6,8], **Aml Abubakr Tantawy**[9], **Mostafa B. Mostafa**[1], **Khaled A. Metwally**[10]

**1** Agricultural Engineering Department, Faculty of Agriculture and Natural Resources, Aswan University, Aswan, Egypt, **2** Soils and Water Department, Faculty of Agriculture, Assiut University, Assiut, Egypt, **3** Department of Agricultural Engineering, Faculty of Agriculture, Omar Al Mukhtar University, Al Bayda, Libya, **4** Department of Electrical Engineering, College of Engineering and Technology, University of Doha for Science and Technology, Doha, Qatar, **5** Agricultural Products Process Engineering Department, Faculty of Agricultural Engineering, Al-Azhar University, Cairo, Egypt, **6** Electrical Engineering Department, Faculty of Energy Engineering, Aswan University, Aswan, Egypt, **7** National Advanced School of Engineering, Universit´e de Yaound´e I, Yaound´e, Cameroon, **8** Department of Communications Technology Engineering, Technical College, Imam Ja'afar Al-Sadiq University, Baghdad, Iraq, **9** Food Science and Technology Department, Faculty of Agriculture and Natural Resources, Aswan University, Aswan, Egypt, **10** Soil and Water Sciences Department, Faculty of Technology and Development, Zagazig University, Zagazig, Egypt

* eutychedan@gmail.com (DEMW); Abdallah_elshawadfy@agr.aswu.edu.eg (AEE)

**Data Availability Statement:** All relevant data are within the manuscript

**Funding:** The author(s) received no specific funding for this work.

## Abstract

The purpose of this study is to investigate how a tracking indirect solar dryer (SD) powered by photovoltaic cells affected the drying kinetics (DK) and thermo-environmental conditions of tomato slices. In this current investigation, three air speeds (1, 1.5, and 2 m/s) are used, as well as three slice thicknesses (ST) (4, 6, and 8 mm) and two SD, one of which is integrated with fixed collector motion (FCM) and another with SD tracking collector motion (TCM). The obtained results showed that the drying time (DT) isn't significantly change with increasing air speeds from 1 to 2 m/s, this may be due to many reasons such as short DT, high temperature inside drying room, and little difference between the exanimated air speeds. When the ST is changed from 4 to 8 mm and maintaining constant air speeds, the DT for FCM and TCM rose by roughly 1.667 and 1.6 times, respectively. In addition, the drying coefficient of the TCM is higher than the FCM due to higher temperature. At 1.5 m/s air speed and 8 mm ST, the maximum values of moisture diffusivity (MD) are $7.15\times10^{-10}$ and $9.30\times10^{-10}$ m$^2$/s for both FCM and TCM systems, respectively. During the study of DK, nine drying models and chose the best based on higher $R^2$ and lower $\chi^2$ and RMSE are used. The findings of the DK analysis revealed that the modified two term II model fit the experimental data of various air speeds well when TF was dried using TCM and FCM systems at varying ST. These findings are based on recorded observations. Where the models' $R^2$ values varied from 0.98005 to 0.99942 for FCM system and varied from 0.99386 to 0.99976 for TCM system. Regarding environmental analysis, it is found that the $CO_2$ mitigation per lifetime is ranged between 5334.9–6795.4 tons for FCM and 6305.7–6323.3 tons for TCM.

**Competing interests:** The authors have declared that no competing interests exist.

## 1. Introduction

Tomato fruit (TF) is a widely consumed vegetable that offers high nutritional value. In addition to being delicious, ripe TF is also a good source of dietary fiber, minerals, vitamins, salt, and natural acids. TF's structural composition includes the following: 0.6–6.6% dry matter, 0.95–1.0% protein, 4.0–5.0% sugar, 0.2–0.3% fat, 0.8–0.9% cellulose, 0.6 ash, 0.5% organic acids, 19–35 mg/kg vitamin C, 0.2–2 mg/kg carotene, 0.3–1.6 mg/kg thiamine, and 1.5–6 mg/kg riboflavin [1–5]. Based on the latest data from the FAO in 2021, the global production of fresh vegetables is estimated at approximately 1.155 billion tons, and TF make up a significant portion of this production, accounting for 16.38% [6–8]. Additionally, the most extensively grown and eaten products worldwide are those derived from agriculture. Nonetheless, agricultural products may have post-harvest losses ranging from 20 to 50% in the time between marketing and consumption. where there are considerable losses for tomato products, totaling about 45.32%, following harvest. The main reason for these losses is that TF has a high-water content—between 93 and 96%. Due to inadequate storage following harvest, the high moisture content (MC) makes it prone to rotting when exposed to the open air, causing significant losses [9–14]. The process of drying, while widely employed in the food industry as a means of preservation, is associated with substantial energy consumption. Specifically, it accounts for approximately 10 to 15% of the industry's total energy expenditure. Given its significant contribution to overall energy consumption, there is a need to explore ways of optimizing the drying process to reduce energy consumption [15, 16]. The initial costs, scarcity, and environmental implications of fossil fuels have brought to the fore the need to explore alternative sources of energy. In this context, renewable energies have emerged as a viable option that can potentially meet the energy demands of the future while ensuring environmental sustainability. These energy sources, which include hydropower, wind, and solar power, among others, have the ability to lessen environmental harm from greenhouse gas emissions while simultaneously reducing reliance on fossil fuels. Therefore, we must understand the significance of renewable energies and work towards their effective integration into the energy mix of the future [17–20]. Sun drying may be an economical method of drying food; however, it is vulnerable to contamination by dust, rodents, insects, and other factors that can lead to the deterioration of the food. On the other hand, the food industry finds solar drying to be a very attractive substitute since it lessens reliance on fossil fuels, shields food from the elements and solar radiation, boosts drying effectiveness, and improves product quality in terms of color and nutrient content. Therefore, producing high-quality dried food items with little chance of contamination or deterioration is a better option for the food sector [4, 21, 22]. By incorporating solar drying systems with other drying techniques, you can take advantage of the benefits of both methods and create a more efficient and cost-effective hybrid system [23–25]. Particularly in tropical and subtropical areas, solar energy can meet the energy requirements for drying. where the favorable climate makes solar energy a desirable choice for drying and where there are good chances that it will be widely adopted [19, 26]. When it comes to solar technology, one of the most important factors in determining if a particular system is appropriate for drying a particular crop is its design for drying systems. In order to guarantee adequate energy efficiency, the design needs to be customized to match the particular needs of the concerned crop. Thus, in order to achieve the best outcomes, solar drying systems must be carefully constructed with attention to every element [21, 27–30].

Auxiliary heating units, energy backup systems, construction materials, design, and structural shape are some of the criteria used to categorize solar dryers. Foods have been dried using a number of sun dryer models. Direct, indirect, mixed-type, and solar dryers with heat storage systems make up the most extensive category of dryers. The primary focus of this

categorization is the process of drying a product by direct or indirect exposure to sun radiation. For food drying applications to function at their best and be as economical and energy-efficient as possible, solar dryers must be properly classified [3, 31–37]. When considering the efficient use of incident solar energy, it was discovered that the northern wall accounts for the majority of the solar fraction loss [38]. Scientists have used many techniques for maximizing the performance of the solar dryers such as; packed bed thermal storage [39], inclinations [40], mirrors [41], phase change materials [42], solar tracking systems [43–45], and parabolic concentrator [46].

The design of drying systems, meeting quality standards, and energy conservation all depend on estimating the overall quality of the dried product and forecasting the food products' DK under varied situations. Where the mathematical representation of the methods of drying is one of the key components of drying technology. There is no denying the significance of process modeling in the design and operation of dryers under ideal drying conditions. Because drying affects the physicochemical and qualitative features of final products, one method for process control is to model the DK [47]. Furthermore, DK facilitates the understanding and quantitative modeling of the thermal and physical variables involved in the drying process [48, 49]. The air temperature, humidity, product size, drying duration, etc. all have a significant impact on the DK [50]. Throughout the drying process, each of them must be taken into consideration since they may all have different effects. This issue renders manual dryer system control all but useless. Finding a model that considers a large number of factors is therefore crucial for researchers. Several drying models have been developed over the past few decades and are commonly used to simulate the DK of food products. Examples of these models include page, Midilli, logistics, etc. [51–54]. In recent years, several researchers have used mathematical models for studying the DK of many agricultural products such as magical berry [55], figs [56], parboiled rice [57], pistachio nuts [58], olives [59], eggplant [60], tomato [61, 62], green peppers [63], strawberries [64], sweet potato and raisins [65], seedless grapes [66], pears [56], green peas [67], pistachio kernels [68], peach [69], orange peel [70], cassava [71], peach slices [72], mango [73], and grapes [74], where they were investigated and researched drying under thin layer (ThL) conditions. The comprehension of the fundamental transport mechanism of materials during the thin-layer drying process is fundamental as it sets the stage for successful simulation or scaling of the entire process to optimize or control operating conditions. Researchers have demonstrated that relying solely on experimental drying techniques without considering the mathematical aspects of drying kinetics can have significant implications on dryer efficiency, increasing production costs, and reducing the quality of the end product. Therefore, an efficient model is indispensable for process design, optimization, energy integration, and control. In this regard, the use of mathematical models to determine the drying kinetics of agricultural products is crucial [75, 76]. This research paper aims to investigate the drying kinetics of both traditional and recently modified methods for drying tomato slices of a renowned local variety using our newly developed solar dryer. As there have been no prior studies conducted on this particular variety, this study will provide suitable data to understand the most appropriate drying model that can enhance the drying process of tomato slices on a commercial scale. The findings of this study will be useful for professionals in the field of food processing who aim to improve the efficiency of their drying methods.

The current study was undertaken to discern the drying performance by utilizing both solar dryers integrated with FCM and TCM systems. By fitting drying curves with well-known models, the best model can be determined. Furthermore, calculate the MD. Finally, an environmental and energy analysis was performed to determine the energy payback time (EPT), and net $CO_2$ mitigation over the lifetime of the developed solar dryer. Through this study, we

hope to add to the body of knowledge already available on the DK of TF varieties and shed light on whether using solar dryers for this purpose would be environmentally feasible.

## 2. Materials and methods

### 2.1. Sample preparation

A fresh TF at an area marketplace in Luxor, Egypt are purchased. On a wet basis (w.b.), the tomato's initial MC was 92%, which is in line with other studies' findings [19, 21, 77]. The selection of TF is based on their consistency in terms of size, color, and maturity. Then, before drying, the chosen TF were cleaned, disinfected, and kept in storage at a consistent 8˚C.

### 2.2. Design of the solar dryers

This study aims to evaluate the DK-based behavior of TF utilizing solar dryers in Luxor City, Egypt. To achieve this, Ref., [16] designed and evaluated a PV-integrated indirect solar dryer incorporating two types of solar collector (SC) (FCM and TCM). We have compared the performance of the two systems for drying thin slices of TF of varying thicknesses (4, 6, and 8 mm) and hot air speeds (1, 1.5, and 2 m/s). Our solar dryer and both SC were designed with a 0.5m × 1.0 m wooden flat plate SC covered with a 3.0 mm glass sheet. The drying room was a 0.44m × 0.63m wooden box with a drying tray covered in plastic mesh and an electronic balance for measuring the periodic weight of the dried TFs. A 40-watt axial flow suction fan operating at a voltage of 220 V was employed to draw ambient air through the SC and into the drying room. An automatic control system for the SC tracker was implemented, comprising an Arduino Uno board, LDR sensors, a linear actuator, a 4-channel relay kit, a photovoltaic (PV) system, a battery, a battery charger, and a measuring unit. The measuring unit is equipped with a DHT-22 sensor to assess ambient temperature and relative humidity. The complete system is represented in Fig 1 according to [16].

### 2.3. Experimental procedure

In August 2023, at Luxor City, tests related to the drying process were carried out. For ten hours every day, starting at seven in the morning and ending at five in the evening, the drying process and data recording were carried out. Every day at 12 p.m., the relative humidity and

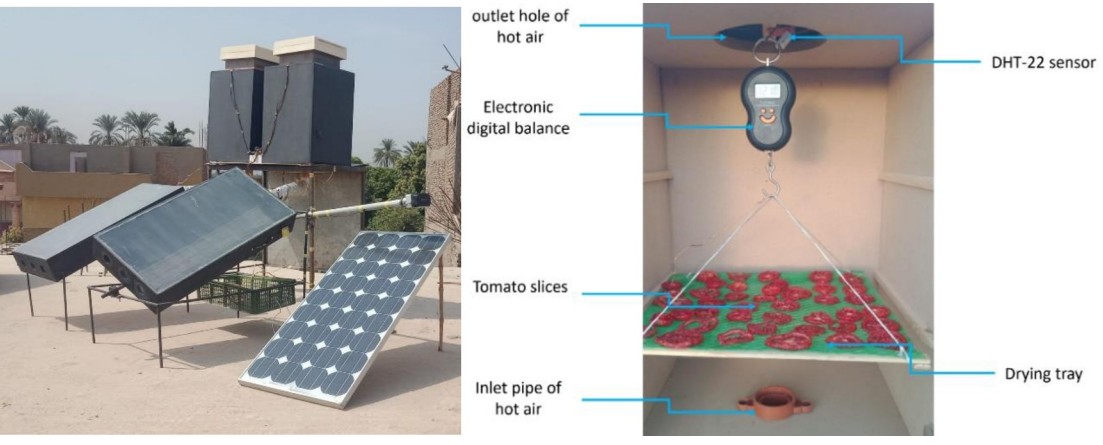

**Fig 1. The SDs integrated with TCM and FCM.**

temperature of the air were recorded. For each variable, the weight of the samples was also measured and recorded daily at 5 p.m.

## 2.4. Calculations and measurements

**2.4.1. Moisture ratio (MR).** Eq 1, as presented in [78], is employed to determine the MR during the drying process. The equation provides a reliable and accurate calculation of the MR, which is essential for further analysis and interpretation of the drying experiments.

$$MR = \frac{M_t - M_e}{M_0 - M_e} \tag{1}$$

where: $M_0$, $M_e$ and $M_t$ are initial MC, equilibrium MC and MC at 't', respectively.

The DK of TF is an area of scientific interest that has been investigated through the use of the MR and corresponding mathematical modeling. It has been determined that $M_e$, in comparison to $M_t$ and $M_0$, exhibits a significantly reduced value, thereby rendering it negligible. As a result, the MR of TF is expressed as shown in Eq 2 as stated in [79].

$$MR = \frac{M}{M_0} \tag{2}$$

**2.4.2. Drying constant ($k$).** The drying constant, or drying coefficient, in the context of ThL drying is derived from a mixture of different drying transport parameters, such as mass coefficients, density, specific heat, MD, thermal conductivity, and interface heat [80]. The drying constant is determined based on the exponential relationship between MR and DT. Furthermore, the determination coefficient is derived from the same relationship for two solar dryers, three ST and three air speeds. While drying constants are vital to comprehensively describing the DK of materials [81, 82], it is crucial to consider the multiple transport properties involved.

$$\frac{dM}{dt} = -k \times (M - M_e) \tag{3}$$

The solution to Eq 10 is obtained through the integration of Eq 4 [83–85],

$$MR = \frac{M_t - M_e}{M_0 - M_e} = \exp(-k \times t) \tag{4}$$

**2.4.3. Drying rate (DR).** The DR is determined using Eq 5, which is reported in [86, 87]. This equation has been widely accepted as a reliable method for calculating the DR and is therefore preferred in academic and industrial settings. The application of this equation ensures that the results obtained are precise and consistent, which is of utmost importance in scientific experimentation.

$$DR = \frac{M_{(t+dt)} - M_t}{d_t} \tag{5}$$

where: $M_t$ is the MC at time 't', while $M_{(t+dt)}$ denotes MC at ($t+dt$), and $d_t$ is the time.

**2.4.4. Moisture diffusivity (MD).** Fick's second law of diffusion, as depicted in Eq 6, has been employed to investigate the drying process during the elimination of moisture, as exemplified in [88]. This diffusion law is a fundamental equation that describes the rate at which a particular species diffuses through a medium, in this case, moisture in a drying process. By

using Fick's second law of diffusion, we can estimate the rate of moisture removal and analyze the drying behavior of materials.

$$\frac{\partial M}{\partial t} = D_{eff} \times \nabla^2 M \tag{6}$$

where: $M$ is the MC, %wb and $t$ is the DT, s.

The drying of food and agricultural commodities, during the falling rate period, can be elucidated through the application of Fick's second law of diffusion. In particular, the answer can be obtained by applying Eq 7, which is created in [89] for an infinite slab, under the assumptions of unidimensional moisture transport, steady temperature, volume change, diffusivity coefficient, and small external resistance. It should be noted that the aforementioned equation [89–93],

$$MR = \frac{M_t}{M_0} = \frac{8}{\pi^2} \times \sum_{n=1}^{\infty} \frac{1}{n^2} exp\left(\frac{-\pi^2 \times D_{eff} \times t}{4L^2}\right) \tag{7}$$

where: $M$ is the final MC; $M_0$ is the initial MC; n is the term number; $D_{eff}$ is the MD in m²/s; $t$ is the time in s; $L$ is the slab thickness (m) from the simplified Ficks diffusion mathematical model.

By discarding the higher-order terms of Eq 7, which are relevant only for longer DT, we can obtain a simplified expression for the MD of TF, as given by Eq 8. This approximation is derived by considering only the first term of the series expansion and neglecting the rest.

$$MR = \frac{8}{\pi^2} \times A \; exp\left(\frac{-\pi^2 \times D_{eff} \times t}{4L^2}\right) \tag{8}$$

Eq 9 has been obtained mathematically by taking the natural logarithm on both sides of Eq 8.

$$ln(MR) = ln\left(\frac{8}{\pi^2}\right) - \left(\frac{\pi^2 \times D_{eff} \times t}{4L^2}\right) \tag{9}$$

The diffusion coefficient is obtained by plotting experimental drying data in terms of $ln(MR)$ versus time, s.

**2.4.5. Mathematical modelling of tomato drying.** Several thin-layer drying models are shown in Table 1 and are employed to evaluate and explain experimental data obtained

Table 1. Mathematical models that explain the drying curve for ThL.

| No. | Model name | Model equation | Reference |
|---|---|---|---|
| 1 | Newton (Lewis) | $MR = exp(-kt)$ | [98] |
| 2 | Page | $MR = exp(-kt^n)$ | [99, 100] |
| 3 | Simplified Ficks Diffusion | $MR = a \exp\left(-c\left(\frac{t}{L^2}\right)\right)$ | [89] |
| 4 | Approximation or diffusion or Diffusion Approach | $MR = a \exp(-kt) + (1-a)exp(-kbt)$ | [101, 102] |
| 5 | Modified Page III | $MR = k \exp\left(-\frac{t}{d^2}\right)^n$ | [75] |
| 6 | Modified Midilli II | $MR = a \exp(-kt^n) + b$ | [75] |
| 7 | Modified Two Term II | $MR = a \exp(-kt) + (1-a)b \; exp(-gt)$ | [75] |
| 8 | Logistics | $MR = \frac{b}{1+a \; \exp(kt)}$ | [103] |
| 9 | Logarithmic | $MR = a \exp(-kt) + c$ | [104] |

* $MR$ is the MR; $L$ is the slab thickness (m); $k$, $k_0$, $k_1$ are the drying constants (day⁻¹); $a$, $b$, $c$, $d$, $g$, $h$, $n$ are the model constants; $t$ is the DT (h).

throughout the drying process. Every drying method's experimental data was fitted to the drying models. Non-linear regression analysis was performed using OriginLab and Microsoft Excel software to estimate the coefficients of the models that were supplied and statistics measures. Where the use of curve fitting to describe experimental data is widespread in all fields of biology. Curve fitting is a method used to standardize data interpretation into a uniformly recognized form. It describes experimental data as a mathematical equation, with the better the fit, the more accurately the function describes the data. Personal computers have reduced the time and effort required for data analysis, making it easier to fit data with simple functions like linear regression. However, fitting data with complex non-linear functions is more difficult and expensive. Specialist programs like Microcal Origin, Sigma Plot, and Graphpad Prism can be expensive and difficult for novices to learn. Microsoft Excel is an alternative method, offering a friendly user interface, flexible data manipulation, built-in mathematical functions, and instantaneous graphing [94]. The best model was selected using the following criteria: the lowest $\chi^2$ and RMSE values, and the greatest $R^2$ [95–97].

These parameters can be calculated using the following Eqs 10–12 according to [78, 105–108],

$$R^2 = 1 - \frac{\sum_{i=1}^{N} \left( MR_{pre,i} - MR_{obs,i} \right)^2}{\sum_{i=1}^{N} \left( \bar{M}R_{pre} - MR_{obs,i} \right)^2} \tag{10}$$

$$\chi^2 = \frac{\sum_{i=1}^{N} \left( MR_{pre,i} - MR_{obs,i} \right)^2}{N - n} \tag{11}$$

$$RMSE = \sqrt{\frac{1}{N} \sum_{i=1}^{N} \left( MR_{pre,i} - MR_{obs,i} \right)^2} \tag{12}$$

where: $MR_{obs,i}$ and $MR_{pre,i}$ are the $i^{th}$ experimental and predicted values; $\bar{M}R_{pre}$ is the average predicted values; $N$ is the number of observations; $n$ is the number of constants in a model [109].

## 2.5. Environmental analysis

**2.5.1. Specific energy consumed (SEC).**   The SEC for drying TF is calculated by the following equation according to [110].

$$SEC = \frac{E_{in}}{M_{out}} \tag{13}$$

where: $E_{in}$ is the input energy to the drying chamber, and $M_{out}$ is the moisture removed from the TF.

**2.5.2. Embodied energy (EE).**   The energy used to manufacture the product is referred to as the EE. All the energy inputs used in the product's creation are taken into account when calculating the coefficient of EE. The coefficient of EE can be multiplied by the product's weight to determine the value of EE [111], where:

$$EE = Coefficeint \; of \; embodied \; energy \; \times \; product \; weight \tag{14}$$

**2.5.3. Energy payback time (EPT).** It is determined using Eq 15 and is defined as the amount of time needed to repay the EE of the developed solar dryer [112, 113]:

$$EPBT = \frac{EE}{Annual\ energy\ output} \tag{15}$$

where: annual energy and daily energy output can be calculated using Eqs 16 and 17 [114, 115]:

$$Annual\ energy\ output = Daily\ energy\ output \times Operating\ days\ /\ year \tag{16}$$

$$Daily\ energy\ output = \frac{m_{ev} \times \lambda}{3.6 \times 10^6} \tag{17}$$

where: $m_{ev}$ is the total water removed from the tomato slices, $\lambda$ is the latent heat of vaporization.

**2.5.4. $CO_2$ emissions.** By taking coal-based power production into account, we can calculate the yearly $CO_2$ emissions of a solar dryer. usually understood to be 0.98 kg/kWh of $CO_2$ [112, 116],

$$CO_2\ emmision\ per\ yaer = \frac{EE \times 0.98}{Lifetime} \tag{18}$$

The following is an expression for Eq 19 that takes into account the losses related to electricity, such as losses related to distribution and transmission ($L_{td}$) and losses from domestic appliances ($L_{da}$).

$$The\ CO_2\ mitigation\ per\ yaer = \frac{1}{1 - L_{da}} \times \frac{1}{1 - L_{td}} \times \frac{EE \times 0.98}{Lifetime} \tag{19}$$

Eq 20 is generated from Eq 20 by assuming the values of $L_{td}$ and $L_{da}$ as 0.4 and 0.2, respectively.

$$The\ CO_2\ mitigation\ per\ yaer = \frac{EE}{Lifetime} \times 2.042\ kg \tag{20}$$

**2.5.5. $CO_2$ mitigation.** The $CO_2$ mitigation per life time of the developed solar dryer can be estimated using Eq 21 as mentioned by [114]:

$$CO_2\ mitigation = [\ Annual\ energy\ output \times Lifetime - EE] \times 2.042\ kg \tag{21}$$

**2.5.6. Carbon credit earned (CCE).** According to [117], CCE is a crucial environmental sustainability measure for renewable energy systems and may be computed using Eq 22.

$$CCE = CO_2\ mitigation \times Cost\ of\ carbon\ credit \tag{22}$$

where the price per ton of $CO_2$ mitigation for CCE ranges from USD 5 to USD 20 [118].

## 2.6. Data measuring (accuracy and error)

Table 2 is an illustration of the accuracy of different devices and sensors utilized in the current study. Where we found that, the total uncertainties in the sensors' reading errors and measurement devices were computed, and the result was ±1.61%. This value is quite small when compared to the acceptable range of ±10% as established by Choi et al. [119]; Rulazi et al. [120].

**Table 2. The Error of the different instruments and sensors.**

| No | Parameters | Unit | Instrument | Range | Accuracy | Resolution | Error, % |
|---|---|---|---|---|---|---|---|
| 1 | Temperature | ˚C | DHT-22 sensor | -10–80˚C | ± 1 | 0.1˚C | 0.1414 |
| 2 | Relative humidity | % | DHT-22 sensor | 0–100% | ± 2 | 0.1% | 0.1414 |
| 3 | Solar radiation | W/m² | Spectral pyranometers | | ± 10 | 0.1 W/m² | 0.1414 |
| 4 | Weight of TF samples | kg | Electronic digital balance | 0.0–50 kg | ± 0.020 | 5 g | 0.707 |
| 5 | Weight of dries TF inside drying room | kg | Electronic digital balance | 0.0–10 kg | ± 10 | 10 g | 1.414 |
| 6 | Weight of dries TF in laboratory | kg | Electronic digital balance | 0.0–1.0 kg | ± 0.15 | 0.1 g | 0.1414 |
| 7 | Voltage and current (PV system) | V, A | Digital multi-meter | 0.2–1000 V 20µA-20A | -- | 0.01 V 0.01 A | 0.01414 0.01414 |
| 8 | Air speed | m/s | A digital anemometer | 0.0–30 m/s | ± 0.1 | 0.1 m/s | 0.1414 |
| 9 | Light intensity | Lux | LDR sensor | 0.0–1000 Lux | ± 1 | 0.1 Lux | 0.1414 |
| 10 | MR* | % | | | | | 0.2 |
| 11 | EE** | kW.h/kg | | | | | 0.707 |

* MR was calculated according to Eq 1, where $M_t$ was measured by instruments no. 5 and 6 in Table 2.

** The EE was calculated according to illustrated data in Table 5, where *the material weight* was measured by instruments no. 4 in Table 2.

## 3. Results and discussions

### 3.1. Moisture ratio

The fluctuation of MR versus DT for solar dryers with FCM and TCM is depicted in Fig 2. Additionally, Fig 2 shows the drying process changes for three different airspeed levels (1, 1.5, and 2 m/s) and three different ST (4, 6, and 8 mm).

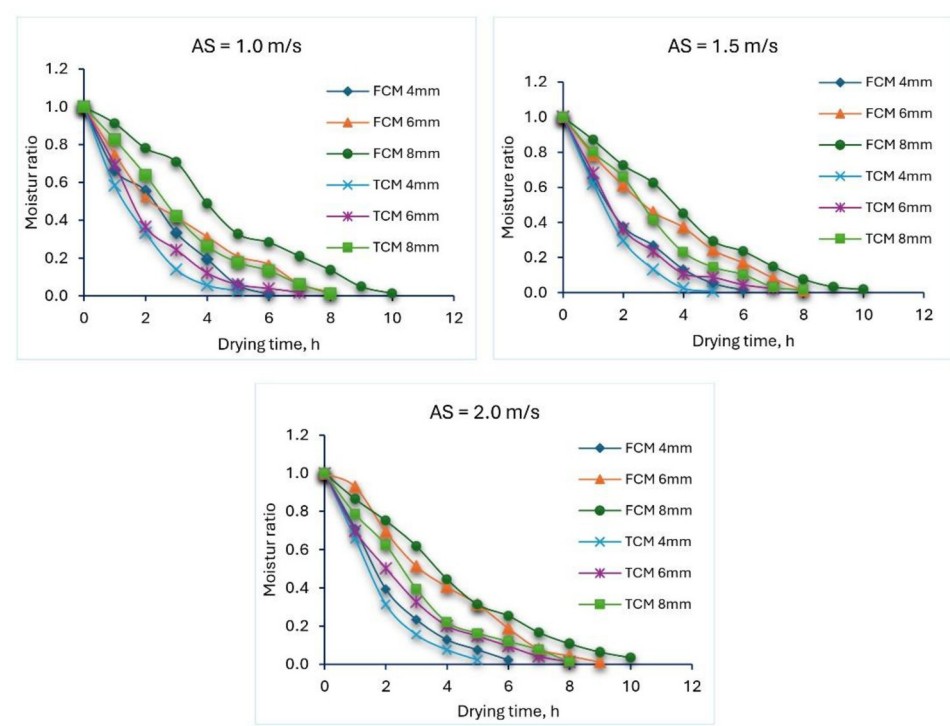

**Fig 2. Drying ratio of different TF samples for both solar dryers at different ST and air speeds.**

**Table 3. Drying coefficient and determination coefficient of TF using solar drying at different air speeds, collector movement and ST.**

| Tomato ST | Coefficient | Air speeds = 1.0 m/s | | Air speeds = 1.5 m/s | | Air speeds = 2.0 m/s | |
|---|---|---|---|---|---|---|---|
| | | FCM | TCM | FCM | TCM | FCM | TCM |
| 4 mm | k | 0.6980 | 0.7390 | 0.6740 | 0.9770 | 0.6080 | 0.7490 |
| | $R^2$ | 0.8868 | 0.9903 | 0.9485 | 0.9625 | 0.9722 | 0.9790 |
| 6 mm | k | 0.4320 | 0.5750 | 0.4620 | 0.5390 | 0.4650 | 0.5030 |
| | $R^2$ | 0.9137 | 0.9945 | 0.8420 | 0.9935 | 0.8948 | 0.9512 |
| 8 mm | k | 0.3780 | 0.4920 | 0.3970 | 0.5170 | 0.3320 | 0.4820 |
| | $R^2$ | 0.8632 | 0.9047 | 0.9309 | 0.9537 | 0.9584 | 0.9053 |

The DR of TF was initially modest at all ST and air speed levels, but it increased as sun radiation increased from 300 to 900 W/m². From 1 to 2 m/s, the DT had no discernible impact on the increasing air speeds. The DT increased approximately by 1.667 to 1.6 times for the FCM and TCM systems, respectively, at constant air speed when the ST was changed from 4 to 8 mm. Where ST had a greater impact on DT than air speed. Where the use of TCM led to a decrease in the DT by about 16.6–36.6% compared with the FCM system. Table 3 lists the drying coefficient (k) and determination coefficient ($R^2$) for the TF at different ST and air speeds. Compared to the air speeds, it is evident that the drying coefficient increases with a decrease in air speeds (air temperature increases) within both solar dryers integrated with FCM and TCM. In addition, the drying coefficient of the TCM was higher than the other one of the FCM due to the higher temperature inside the TCM compared with the FCM. These findings come in agreement with [121–123].

## 3.2. Drying rate (DR)

The DR of different TF samples for both solar dryers at different ST and air speeds is presented in Fig 3. Where the plotted date in the same figure showed that the DR was maximum at the

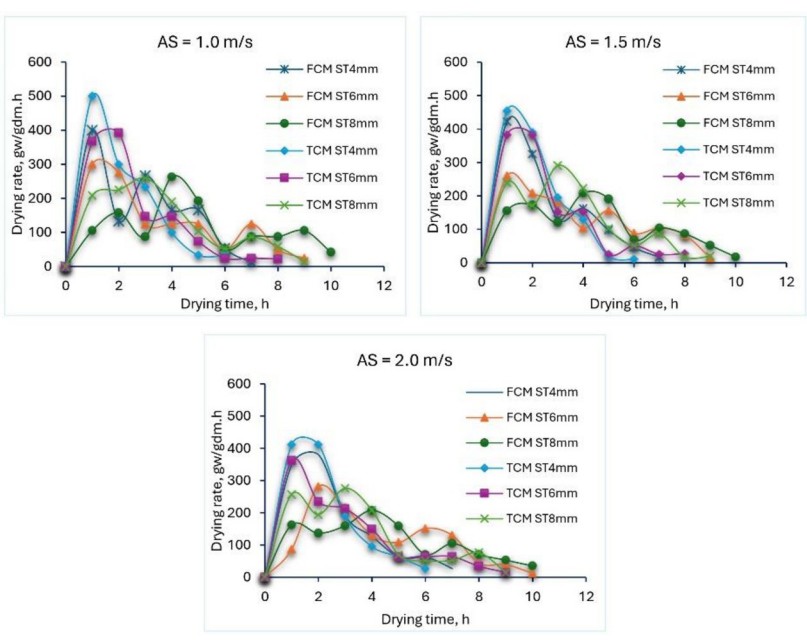

**Fig 3. DR of different TF samples for both solar dryers at different ST and air speeds.**

beginning of the drying process and then decreased gradually, this means that the major value of MC loosed in the falling rate phase, which is consistent with other studies that other researchers have observed [51, 124–136].

The maximum DR was recorded with an ST of 4 mm, followed by an ST of 6 mm, while the lowest DR was observed with an ST of 8 mm. In addition, the use of TCM led to an increase in DR values at all ST and air speeds compared to FCM. While we found that there was a slight difference in DR at different air speeds during the current study.

## 3.3. Moisture diffusion (MD)

Figs 4 and 5 show the diffusivity of TF using solar drying at different air speeds, collector movement, and ST. The presented data shows that the MD did not take a constant trend at different air speeds; on the other hand, the values of MD increased with increasing the ST for both systems. Because of the increased thermal energy inside the collection and dryer cabinet, the MD values for the TCM system were higher than those for the FCM system at the same levels of air speeds and ST. These findings are consistent in [93]. The maximum MD was

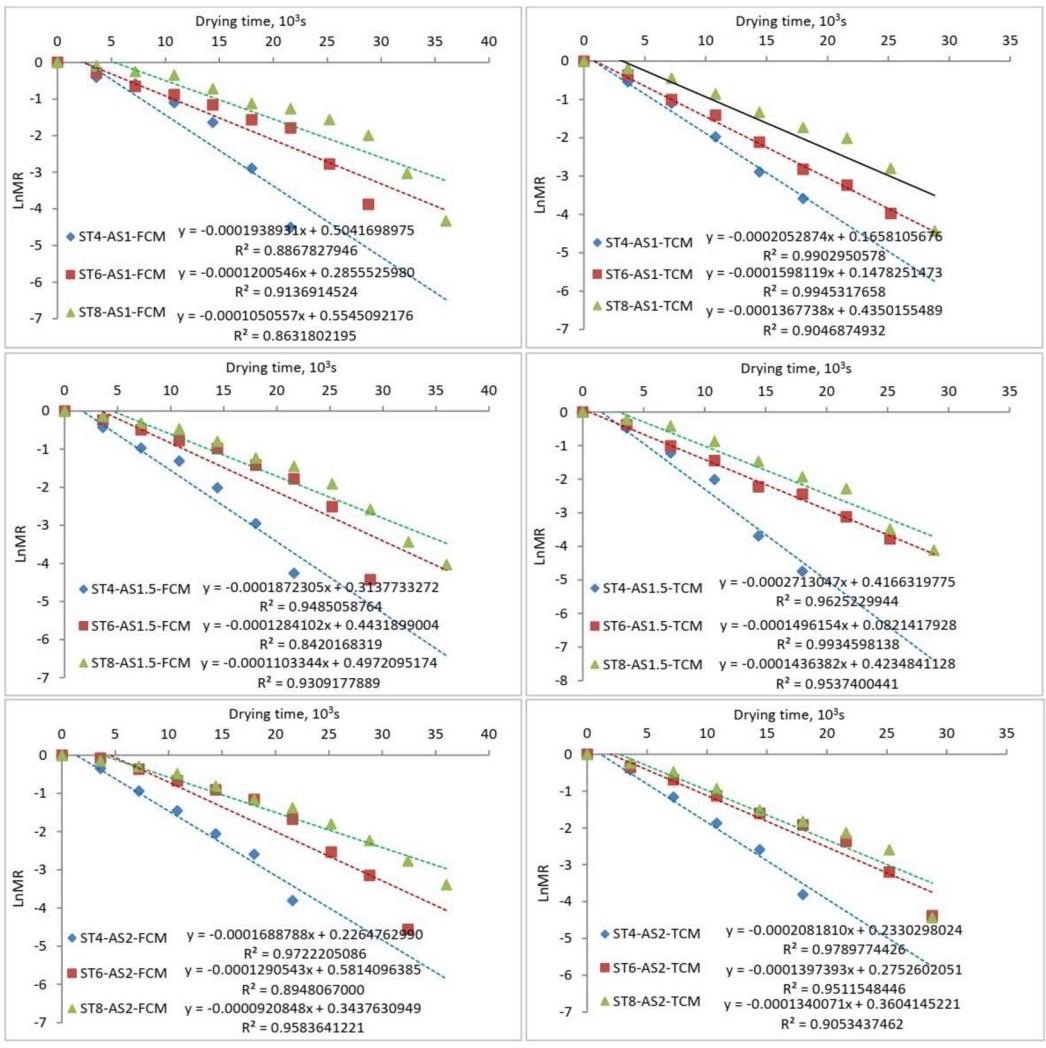

**Fig 4. Diffusivity of TF using solar drying at different ST, air speeds, and collector movement.**

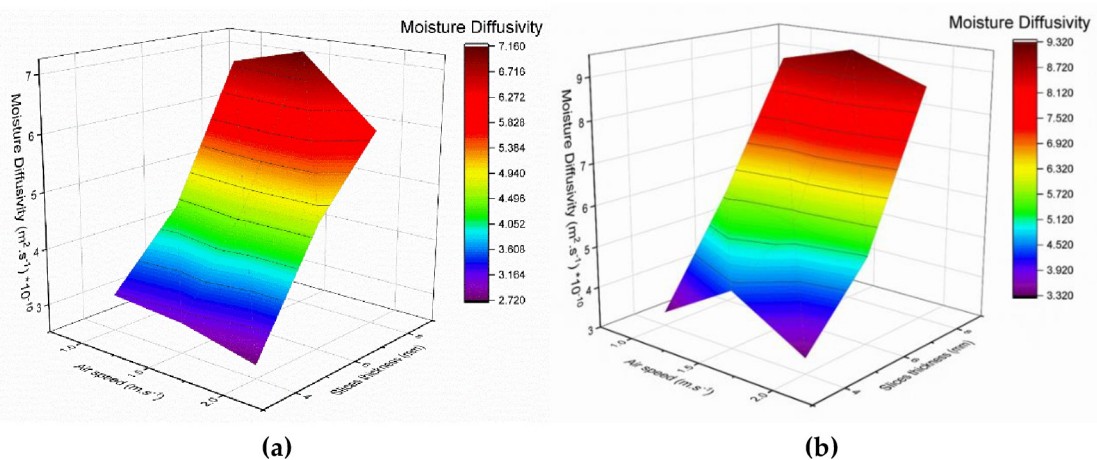

**Fig 5. Diffusivity of TF using solar drying at different air speeds, collector movement and ST, a: TCM system and b: FCM system.**

$7.15 \times 10^{-10}$ and $9.30 \times 10^{-10}$ m$^2$/s for both FCM and TCM systems, respectively, at air speeds of 1.5 m/s and ST of 8 mm. While the minimum MD was $2.73 \times 10^{-10}$ m$^2$/s for FCM at air speeds of 2 m/s and ST of 4 mm and $3.32 \times 10^{-10}$ m$^2$/s for FCM at 1 m/s air speeds and 4 mm ST.

### 3.4. Drying kinetics (DK)

Table 4 shows a statistical analysis of different ThL models of TF at different ST and air speeds. Different standard ThL models were used to study the DK. At the beginning, the MC data was collected for TF at different ST and air speeds. After that, the MC was converted into the MR expression, and nine ThL models were used to compute curve fitting. The findings of the statistical analysis, which are shown in Table 4, show that all DK had an overall high R$^2$, low $\chi^2$, and RMSE, which are examples of statistical measures that were used to assess the quality of the fitted models. Many researchers demonstrated that the model most suited for defining the ThL drying was the one with the greatest R$^2$, lowest $\chi^2$, and RMSE values [95–97]. As shown in Table 4, the Modified Two-Term II model was found to exhibit a good fit to the experimental data of different air speeds for drying TF at different ST using TCM and FCM systems. These findings were based on recorded observations. The models' R$^2$ values varied from 0.98005 to 0.99942 for the FCM system and from 0.99386 to 0.99976 for the TCM system.

### 3.5. Thermo-environmental analysis

It is very important to do thermo-Environmental analysis for both solar dryers integrated with FCM and TCM. Table 5 demonstrates the EE of the different material used in manufacturing the solar dryer integrated with TCM and PV system. The total EE of the solar dryer, collector and PV system was 1140.8 kW.h as shown in Table 5. Because there are not significant differences between the air speeds during the current study, the thermo-environmental analysis was conducted based on the three ST and collector movement (FCM and TCM). The tabulated data in Table 5, showed that the SEC was calculated based on total solar radiations fall on the solar collector during day and amount of water evaporative from the tomato slices. Based on the obtained data, we noticed there are reverse relation between the SEC and ST, where the SEC was increasing with increasing the ST, in addition maximum SEC values was observed with TCM comparing with FCM. Where maximum SEC was 8.15 and 5.42 kW.h/kg for FCM and TCM at 8.0 mm ST.

**Table 4. Statistical analysis of different ThL models of TF.**

| Model name | AS, m/s | Tomato ST, mm | FCM | | | | TCM | | | |
|---|---|---|---|---|---|---|---|---|---|---|
| | | | Model constants | $R^2$ | $\chi^2$ | RMSE | Model constants | $R^2$ | $\chi^2$ | RMSE |
| **Newton (Lewis)** | 1.0 | 4.0 | k = 0.39520 | 0.96762 | 0.00417 | 0.05978 | k = 0.59425 | 0.993643 | 0.000922 | 0.027720 |
| | | 6.0 | k = 0.31401 | 0.99032 | 0.00103 | 0.03027 | k = 0.48019 | 0.99093 | 0.00115 | 0.03171 |
| | | 8.0 | k = 0.20236 | 0.93283 | 0.00836 | 0.08717 | k = 0.30034 | 0.97635 | 0.00270 | 0.04928 |
| | 1.5 | 4.0 | k = 0.48153 | 0.99305 | 0.00089 | 0.02759 | k = 0.61129 | 0.90400 | 0.00242 | 0.04492 |
| | | 6.0 | k = 0.28071 | 0.97980 | 0.00223 | 0.04456 | k = 0.48419 | 0.99296 | 0.00087 | 0.02761 |
| | | 8.0 | k = 0.22929 | 0.95228 | 0.00586 | 0.07301 | k = 0.32197 | 0.96128 | 0.00508 | 0.06719 |
| | 2.0 | 4.0 | k = 0.46664 | 0.98796 | 0.00160 | 0.03704 | k = 0.56200 | 0.98516 | 0.00219 | 0.04268 |
| | | 6.0 | k = 0.24889 | 0.94773 | 0.00679 | 0.07817 | k = 0.38054 | 0.99564 | 0.00049 | 0.02092 |
| | | 8.0 | k = 0.22027 | 0.96117 | 0.00451 | 0.06404 | k = 0.32480 | 0.97576 | 0.00296 | 0.05132 |
| **Page** | 1.0 | 4.0 | k = 0.29436 n = 1.26807 | 0.97982 | 0.00312 | 0.04719 | k = 0.51637 n = 1.19048 | 0.99852 | 0.00027 | 0.01340 |
| | | 6.0 | k = 0.27934 n = 1.08920 | 0.99230 | 0.00094 | 0.02699 | k = 0.39483 n = 1.21206 | 0.99774 | 0.00029 | 0.01590 |
| | | 8.0 | k = 0.07006 n = 1.64585 | 0.99228 | 0.00107 | 0.02955 | k = 0.18623 n = 1.37429 | 0.99791 | 0.00027 | 0.01460 |
| | 1.5 | 4.0 | k = 0.42637 n = 1.13006 | 0.99614 | 0.00059 | 0.02058 | k = 0.47223 n = 1.36839 | 0.99566 | 0.00014 | 0.00955 |
| | | 6.0 | k = 0.20397 n = 1.23026 | 0.99117 | 0.00098 | 0.02947 | k = 0.41542 n = 1.16817 | 0.99735 | 0.00038 | 0.01695 |
| | | 8.0 | k = 0.09836 n = 1.53630 | 0.99553 | 0.00061 | 0.02234 | k = 0.17076 n = 1.49998 | 0.99550 | 0.00067 | 0.02290 |
| | 2.0 | 4.0 | k = 0.36665 n = 1.26163 | 0.99863 | 0.00022 | 0.01248 | k = 0.43744 n = 1.33683 | 0.99910 | 0.00017 | 0.01052 |
| | | 6.0 | k = 0.11205 n = 1.53803 | 0.99317 | 0.00100 | 0.02825 | k = 0.33395 n = 1.11224 | 0.99826 | 0.00023 | 0.01323 |
| | | 8.0 | k = 0.10497 n = 1.46532 | 0.99750 | 0.00032 | 0.01626 | k = 0.21171 n = 1.34047 | 0.99516 | 0.00068 | 0.02294 |
| **Simplified Ficks Diffusion** | 1.0 | 4.0 | a = 1.02234 c = 0.76516 L = 1.37707 | 0.96846 | 0.00609 | 0.05901 | a = 1.01470 c = 0.86413 L = 1.19830 | 0.99397 | 0.00146 | 0.02699 |
| | | 6.0 | a = 1.01276 c = 0.69402 L = 1.47727 | 0.99059 | 0.00134 | 0.02985 | a = 1.02819 c = 0.81661 L = 1.28810 | 0.99199 | 0.00142 | 0.02979 |
| | | 8.0 | a = 1.10644 c = 0.62508 L = 1.67077 | 0.94886 | 0.00797 | 0.07612 | a = 1.06082 c = 0.69661 L = 1.47949 | 0.98120 | 0.00272 | 0.04364 |
| | 1.5 | 4.0 | a = 1.01440 c = 0.81342 L = 1.29123 | 0.99337 | 0.00170 | 0.02695 | a = 1.02804 c = 0.87847 L = 1.18530 | 0.91098 | 0.00374 | 0.04326 |
| | | 6.0 | a = 1.03281 c = 0.67423 L = 1.52485 | 0.98159 | 0.00272 | 0.04255 | a = 1.02338 c = 0.81602 L = 1.28478 | 0.99370 | 0.00109 | 0.02612 |
| | | 8.0 | a = 1.08529 c = 0.64959 L = 1.61842 | 0.96218 | 0.00582 | 0.06506 | a = 1.06770 c = 0.71638 L = 1.44785 | 0.96745 | 0.00570 | 0.06164 |
| | 2.0 | 4.0 | a = 1.03292 c = 0.81402 L = 1.30160 | 0.98958 | 0.00208 | 0.03446 | a = 1.03006 c = 0.85289 L = 1.21632 | 0.98656 | 0.00495 | 0.04063 |
| | | 6.0 | a = 1.09914 c = 0.66743 L = 1.56507 | 0.96102 | 0.00652 | 0.06755 | a = 1.01543 c = 0.74703 L = 1.39103 | 0.99598 | 0.00061 | 0.02009 |
| | | 8.0 | a = 1.07857 c = 0.64097 L = 1.64308 | 0.97012 | 0.00434 | 0.05621 | a = 1.05198 c = 0.71309 L = 1.44688 | 0.97960 | 0.00333 | 0.04710 |
| **Approximation or diffusion or Diffusion Approach** | 1.0 | 4.0 | k = 0.39520 a = 1.00000 b = 1.00000 | 0.96762 | 0.00625 | 0.05978 | k = 0.59425 a = 1.00000 b = 1.00000 | 0.99364 | 0.00154 | 0.02772 |
| | | 6.0 | k = 0.31401 a = 1.00000 b = 1.00000 | 0.99032 | 0.00137 | 0.03027 | k = 0.48019 a = 1.00000 b = 1.00000 | 0.99093 | 0.00161 | 0.03171 |
| | | 8.0 | k = 0.20236 a = 1.00000 b = 1.00000 | 0.93283 | 0.01045 | 0.08717 | k = 0.30034 a = 1.00000 b = 1.00000 | 0.97614 | 0.00347 | 0.04928 |
| **Approximation or diffusion or Diffusion Approach** | 1.5 | 4.0 | k = 0.48153 a = 1.00000 b = 1.00000 | 0.99305 | 0.00133 | 0.02759 | k = 0.61129 a = 1.00000 b = 1.00000 | 0.90400 | 0.00404 | 0.04492 |
| | | 6.0 | k = 0.28071 a = 1.00000 b = 1.00000 | 0.97980 | 0.00298 | 0.04456 | k = 0.48419 a = 1.00000 b = 1.00000 | 0.99296 | 0.00122 | 0.02761 |
| | | 8.0 | k = 0.22929 a = 1.00000 b = 1.00000 | 0.95228 | 0.00733 | 0.07301 | k = 0.32197 a = 1.00000 b = 1.00000 | 0.96128 | 0.00677 | 0.06719 |
| | 2.0 | 4.0 | k = 0.46664 a = 1.00000 b = 1.00000 | 0.98796 | 0.00240 | 0.03704 | k = 0.56200 a = 1.00000 b = 1.00000 | 0.98516 | 0.00364 | 0.04268 |
| | | 6.0 | k = 0.24889 a = 1.00000 b = 1.00000 | 0.94773 | 0.00873 | 0.07817 | k = 0.38054 a = 1.00000 b = 1.00000 | 0.99564 | 0.00066 | 0.02092 |
| | | 8.0 | k = 0.22027 a = 1.00000 b = 1.00000 | 0.96117 | 0.00564 | 0.06404 | k = 0.32480 a = 1.00000 b = 1.00000 | 0.97576 | 0.00395 | 0.05132 |

(*Continued*)

**Table 4.** (Continued)

| Model name | AS, m/s | Tomato ST, mm | FCM | | | | TCM | | | |
|---|---|---|---|---|---|---|---|---|---|---|
| | | | Model constants | R² | χ² | RMSE | Model constants | R² | χ² | RMSE |
| **Modified Page III** | 1.0 | 4.0 | k = 1.02234 d = 1.37707 n = 0.76516 | 0.96846 | 0.00609 | 0.05901 | k = 1.01470 d = 1.19830 n = 0.86413 | 0.99397 | 0.00146 | 0.02699 |
| | | 6.0 | k = 1.01276 d = 1.47727 n = 0.69402 | 0.99059 | 0.00134 | 0.02985 | k = 1.02819 d = 1.28810 n = 0.81661 | 0.99199 | 0.00142 | 0.02979 |
| | | 8.0 | k = 1.10644 d = 1.67077 n = 0.62508 | 0.94886 | 0.00797 | 0.07612 | k = 1.06082 d = 1.47949 n = 0.69661 | 0.98120 | 0.00272 | 0.04364 |
| | 1.5 | 4.0 | k = 1.01440 d = 1.29123 n = 0.81342 | 0.99337 | 0.00127 | 0.02695 | k = 1.02804 d = 1.18530 n = 0.87847 | 0.91098 | 0.00374 | 0.04326 |
| | | 6.0 | k = 1.03281 d = 1.52485 n = 0.67423 | 0.98159 | 0.00272 | 0.04255 | k = 1.02338 d = 1.28478 n = 0.81602 | 0.99370 | 0.00109 | 0.02612 |
| | | 8.0 | k = 1.08529 d = 1.61842 n = 0.64959 | 0.96218 | 0.00582 | 0.06506 | k = 1.06770 d = 1.44785 n = 0.71638 | 0.96745 | 0.00570 | 0.06164 |
| | 2.0 | 4.0 | k = 1.03292 d = 1.30160 n = 0.81402 | 0.98958 | 0.00208 | 0.03446 | k = 1.03006 d = 1.21632 n = 0.85289 | 0.98656 | 0.00330 | 0.04063 |
| | | 6.0 | k = 1.09914 d = 1.56507 n = 0.66743 | 0.96102 | 0.00652 | 0.06755 | k = 1.01543 d = 1.39103 n = 0.74703 | 0.99598 | 0.00061 | 0.02009 |
| | | 8.0 | k = 1.07857 d = 1.64308 n = 0.64097 | 0.97012 | 0.00434 | 0.05621 | k = 1.05198 d = 1.44688 n = 0.71309 | 0.97960 | 0.00333 | 0.04710 |
| **Modified Midilli II** | 1.0 | 4.0 | k = 0.27026 a = 0.97456 b = 0.00000 n = 1.32003 | 0.98060 | 0.00499 | 0.04626 | k = 0.51405 a = 0.99758 b = 0.00000 n = 1.19343 | 0.99853 | 0.00054 | 0.01336 |
| | | 6.0 | k = 0.27398 a = 0.99340 b = 0.00000 n = 1.09863 | 0.99236 | 0.00130 | 0.02690 | k = 0.42046 a = 1.00776 b = 0.00000 n = 1.16848 | 0.99744 | 0.00057 | 0.01686 |
| | | 8.0 | k = 0.06539 a = 0.98856 b = 0.00000 n = 1.67896 | 0.99243 | 0.00135 | 0.02926 | k = 0.18289 a = 0.98266 b = 0.01654 n = 1.41986 | 0.99811 | 0.00032 | 0.01388 |
| | 1.5 | 4.0 | k = 0.42411 a = 0.99751 b = 0.00000 n = 1.13311 | 0.99614 | 0.00074 | 0.02055 | k = 0.47141 a = 0.99913 b = 0.00000 n = 1.36965 | 0.99567 | 0.00027 | 0.00954 |
| | | 6.0 | k = 0.19079 a = 0.98261 b = 0.00000 n = 1.26398 | 0.99152 | 0.00150 | 0.02887 | k = 0.44266 a = 0.98986 b = 0.01973 n = 1.18148 | 0.99763 | 0.00051 | 0.01600 |
| | 1.5 | 8.0 | k = 0.08639 a = 0.97666 b = 0.00000 n = 1.60056 | 0.99609 | 0.00069 | 0.02091 | k = 0.16123 a = 0.98640 b = 0.00000 n = 1.53312 | 0.99570 | 0.00090 | 0.02240 |
| **Modified Midilli II** | 2.0 | 4.0 | k = 0.40318 a = 1.01789 b = 0.00000 n = 1.16817 | 0.99768 | 0.00062 | 0.01627 | k = 0.45658 a = 1.00830 b = 0.00000 n = 1.29718 | 0.99896 | 0.00038 | 0.01130 |
| | | 6.0 | k = 0.11539 a = 1.00632 b = 0.00000 n = 1.52327 | 0.99322 | 0.00132 | 0.02816 | k = 0.32996 a = 0.99529 b = 0.00000 n = 1.11848 | 0.99828 | 0.00031 | 0.01314 |
| | | 8.0 | k = 0.09672 a = 0.98430 b = 0.00000 n = 1.50469 | 0.99776 | 0.00037 | 0.01537 | k = 0.20926 a = 0.99219 b = 0.00351 n = 1.35317 | 0.99519 | 0.00094 | 0.02286 |
| **Modified Two Term II** | 1.0 | 4.0 | k = 0.63266 a = 5.06647 b = 1.00331 g = 0.73277 | 0.98005 | 0.00514 | 0.04692 | k = 0.88536 a = 3.75923 b = 1.00033 g = 1.05955 | 0.99856 | 0.00052 | 0.01321 |
| | | 6.0 | k = 0.39992 a = 1.99865 b = 1.00241 g = 0.53126 | 0.99265 | 0.00125 | 0.02638 | k = 0.57597 a = 1.22505 b = 0.99998 g = 12.63889 | 0.99862 | 0.00031 | 0.01238 |
| | | 8.0 | k = 0.39842 a = 4.34618 b = 1.00456 g = 0.53314 | 0.98913 | 0.00193 | 0.03507 | k = 0.46196 a = 1.95147 b = 1.00309 g = 0.89402 | 0.99819 | 0.00031 | 0.01356 |
| | 1.5 | 4.0 | k = 0.65493 a = 2.56432 b = 1.00068 g = 0.82904 | 0.99622 | 0.00097 | 0.02036 | k = 1.05336 a = 4.38293 b = 1.00027 g = 1.31859 | 0.99386 | 0.00039 | 0.01136 |
| | | 6.0 | k = 0.43541 a = 3.44784 b = 1.00442 g = 0.54570 | 0.99138 | 0.00153 | 0.02912 | k = 0.56702 a = 1.18969 b = 0.99923 g = 10.20253 | 0.99849 | 0.00033 | 0.01279 |
| | | 8.0 | k = 0.42850 a = 4.62554 b = 1.00462 g = 0.54717 | 0.99266 | 0.00150 | 0.02864 | k = 0.58591 a = 4.29334 b = 1.00300 g = 0.75953 | 0.99394 | 0.00127 | 0.02658 |
| | 2.0 | 4.0 | k = 0.57498 a = 1.25940 b = 0.99913 g = 7.03240 | 0.99942 | 0.00015 | 0.00813 | k = 0.74712 a = 1.40591 b = 0.99997 g = 3.72163 | 0.99976 | 0.00009 | 0.00541 |
| | | 6.0 | k = 0.45944 a = 3.58694 b = 0.99989 g = 0.64672 | 0.99189 | 0.00158 | 0.03079 | k = 0.49078 a = 1.86667 b = 1.00520 g = 0.70430 | 0.99843 | 0.00028 | 0.01257 |
| | | 8.0 | k = 0.39188 a = 3.47861 b = 1.00540 g = 0.54106 | 0.99622 | 0.00063 | 0.01999 | k = 0.51506 a = 2.48243 b = 1.00416 g = 0.79588 | 0.99513 | 0.00095 | 0.02299 |

(*Continued*)

**Table 4.** (Continued)

| Model name | AS, m/s | Tomato ST, mm | FCM | | | | TCM | | | |
|---|---|---|---|---|---|---|---|---|---|---|
| | | | Model constants | $R^2$ | $\chi^2$ | RMSE | Model constants | $R^2$ | $\chi^2$ | RMSE |
| Logistics | 1.0 | 4.0 | k = 0.67074 a = 0.48386 b = 1.44260 | 0.98337 | 0.00321 | 0.04284 | k = 0.83226 a = 1.07323 b = 2.06768 | 0.99873 | 0.00031 | 0.01242 |
| | | 6.0 | k = 0.38492 a = 2.13755b = 3.11640 | 0.99264 | 0.00105 | 0.02639 | k = 0.68786 a = 1.02291 b = 2.03386 | 0.99736 | 0.00047 | 0.01711 |
| | | 8.0 | k = 0.49525 a = 0.18079 b = 1.18259 | 0.99242 | 0.00118 | 0.02929 | k = 0.55657 a = 0.43806 b = 1.44792 | 0.99765 | 0.00034 | 0.01546 |
| | 1.5 | 4.0 | k = 0.61536 a = 1.68264 b = 2.67677 | 0.99620 | 0.00073 | 0.02041 | k = 1.06370 a = 0.46911 b = 1.46825 | 0.99523 | 0.00020 | 0.01002 |
| | | 6.0 | k = 0.44469 a = 0.67099 b = 1.64497 | 0.99274 | 0.00107 | 0.02671 | k = 0.65000 a = 1.38085 b = 2.39488 | 0.99705 | 0.00051 | 0.01786 |
| | 1.5 | 8.0 | k = 0.51254 a = 0.21662 b = 1.20049 | 0.99716 | 0.00044 | 0.01782 | k = 0.68870 a = 0.24597 b = 1.23479 | 0.99684 | 0.00055 | 0.01919 |
| Logistics | 2.0 | 4.0 | k = 0.72173 a = 0.76764 b = 1.77722 | 0.99818 | 0.00036 | 0.01439 | k = 0.95712 a = 0.53275 b = 1.53750 | 0.99885 | 0.00042 | 0.01187 |
| | | 6.0 | k = 0.52680 a = 0.28107 b = 1.30079 | 0.99193 | 0.00135 | 0.03071 | k = 0.47967 a = 1.80169 b = 2.78896 | 0.99849 | 0.00017 | 0.01231 |
| | | 8.0 | k = 0.45846 a = 0.28672 b = 1.27905 | 0.99839 | 0.00023 | 0.01303 | k = 0.57641 a = 0.46960 b = 1.46811 | 0.99554 | 0.00073 | 0.02202 |
| Logarithmic | 1.0 | 4.0 | k = 0.40353 a = 1.02238 c = 0.00000 | 0.96846 | 0.00609 | 0.05901 | k = 0.60179 a = 1.01470 c = 0.00000 | 0.99397 | 0.00146 | 0.02699 |
| | | 6.0 | k = 0.31802 a = 1.01276 c = 0.00000 | 0.99059 | 0.00134 | 0.02985 | k = 0.49215 a = 1.02817 c = 0.00000 | 0.99199 | 0.00142 | 0.02979 |
| | | 8.0 | k = 0.22393 a = 1.10643 c = 0.00000 | 0.94886 | 0.00797 | 0.07612 | k = 0.31825 a = 1.06082 c = 0.00000 | 0.98120 | 0.00272 | 0.04364 |
| | 1.5 | 4.0 | k = 0.48787 a = 1.01440 c = 0.00000 | 0.99337 | 0.00127 | 0.02695 | k = 0.62527 a = 1.02804 c = 0.00000 | 0.91098 | 0.00374 | 0.04326 |
| | | 6.0 | k = 0.28997 a = 1.03281 c = 0.00000 | 0.98159 | 0.00204 | 0.04255 | k = 0.49437 a = 1.02338 c = 0.00000 | 0.99370 | 0.00109 | 0.02612 |
| | | 8.0 | k = 0.24800 a = 1.08529 c = 0.00000 | 0.96218 | 0.00582 | 0.06506 | k = 0.34174 a = 1.06770 c = 0.00000 | 0.96745 | 0.00427 | 0.06164 |
| | 2.0 | 4.0 | k = 0.48045 a = 1.03287 c = 0.00000 | 0.98958 | 0.00208 | 0.03446 | k = 0.57650 a = 1.03006 c = 0.00000 | 0.98656 | 0.00330 | 0.04063 |
| | | 6.0 | k = 0.27248 a = 1.09914 c = 0.00000 | 0.96102 | 0.00652 | 0.06755 | k = 0.38607 a = 1.01543 c = 0.00000 | 0.99598 | 0.00061 | 0.02009 |
| | | 8.0 | k = 0.23742 a = 1.07857 c = 0.00000 | 0.97012 | 0.00434 | 0.05621 | k = 0.34062 a = 1.05198 c = 0.00000 | 0.97960 | 0.00333 | 0.04710 |

Also, annual thermal output of both solar dryer and PV system was calculated and showed that values was 375.34, 399.23, and 446.86 kW.h for the solar dryer integrated with FCM, and was 423.74, 422.88, and 474.85 kW.h for the solar dryer integrated with TCM at ST of 8, 6, and 4 mm, respectively. Small ST and high temperature inside the drying room of the solar dryer integrated with TCM led to evaporate more quantities of water from tomato slices compared to FCM at the same time, in addition to FCM led to dry the tomato slices shorter than FCM that can lead to dry more quantities of tomato slices at the same time. Maximum EPT calculated and found to be 3.04 and 2.69 year for both FCM and TCM at 8 mm ST, respectively. The EPT was decreasing with decreasing the ST where the lowest EPT was observed at 4 mm ST. The $CO_2$ emission from the solar dryer was 137.11 tons while $CO_2$ mitigation per lifetime 5334.9, 5822.8 and 6795.4 tons for FCM and 6323.3, 6305.7 and 7366.9 tons for TCM. Minimum CCE of the solar dryer integrated with FCM and TCM was (26674, 29114 and 33977 $), and (31616, 31528 and 36834 $), respectively. The obtained results in Table 6 are in agreement with [20, 139]. The scientists and researchers working in the subject of sun drying would find this data interesting, and the above-developed approaches can be applied to various commercial-scale solar drying system designs [20, 140]. Environmental metrics of performance show

**Table 5. EE of the solar dryer integrated with TCM.**

| No | Materials | EE, kW.h/kg | Weight kg | EE (kW.h) | References |
|---|---|---|---|---|---|
| | | Solar collector | | | |
| 1 | Wooden frame | 2.0 | 18.0 | 36.0 | [112, 137] |
| 2 | Glass cover | 7.28 | 8.0 | 58.24 | [138] |
| 4 | Paint | 25.11 | 0.5 | 12.56 | |
| 5 | Absorber plate | 4.20 | 9.0 | 37.8 | |
| | | Drying room | | | |
| 1 | Wooden frame | 2.0 | 10 | 20.0 | [112, 137] |
| 4 | Hinges | 55.28 | 0.05 | 2.76 | [138] |
| | Handel | 55.28 | 0.05 | 7.26 | |
| 5 | *Suction fan* | | | | |
| | 1. Plastic parts | 19.44 | 0.20 | 3.89 | |
| | 2. Motor and cooper wires | 19.6 | 0.20 | 3.92 | |
| 6 | *Drying trays* | | | | |
| | 1. Plastic mesh | 19.44 | 0.5 | 9.72 | |
| | 2. Wooden frame | 2.0 | 5.0 | 10 | |
| Total EE for solar dryer (solar collector + drying room) (kWh), 202.15 | | | | | |
| | | PV system | | | |
| 1 | Metal frame | 55.28 | 1.0 | 55.28 | [112, 137] |
| 2 | PV system | 1130.6 W.h/m$^2$ | 0.65 m$^2$ | 734.89 | |
| 3 | Battery | 148.4515 | -- | 148.45 | |
| 4 | Battery charger | -- | -- | | |
| Total EE for PV system (kWh) | | | | 938.62 | |

how the designed sun-drying system affects the environment and global warming. Table 7 shows a comparison between energy payback time of our model with other similar models.

## 4. Conclusion and future work

The poor performance of solar dryers has long been a source of criticism. The quality of the dried product, environmental factors, DK, thermal performance, and other factors can all be

**Table 6. Analysis of different environmental parameters.**

| Environmental parameters | Collector type | Tomato ST, mm | | |
|---|---|---|---|---|
| | | **8.0** | **6.0** | **4.0** |
| Specific energy consumed (SEC), kW.h/kg | FCM | 8.15 | 6.53 | 4.69 |
| | TCM | 5.42 | 4.46 | 4.01 |
| Embodied energy (EE), kW.h | FCM | 1140.8 | 1140.8 | 1140.8 |
| | TCM | 1140.8 | 1140.8 | 1140.8 |
| Total annual energy output, kW.h | FCM | 375.34 | 399.23 | 446.86 |
| | TCM | 423.74 | 422.88 | 474.85 |
| Energy payback time (EPT), year | FCM | 3.04 | 2.86 | 2.55 |
| | TCM | 2.69 | 2.70 | 2.40 |
| $CO_2$ emissions, ton | FCM | 137.11 | 137.11 | 137.11 |
| | TCM | 137.11 | 137.11 | 137.11 |
| $CO_2$ mitigation per lifetime, ton | FCM | 5334.9 | 5822.8 | 6795.4 |
| | TCM | 6323.3 | 6305.7 | 7366.9 |
| Minimum carbon credit earned, USD | FCM | 26674 | 29114 | 33977 |
| | TCM | 31616 | 31528 | 36834 |

**Table 7. Comparison between energy payback time of our model with other similar models.**

| Reference | Publisher | year | Type of the solar dryer | Dried product | Energy payback time |
|---|---|---|---|---|---|
| Brahma et al. [141] | Elsevier | 2024 | Novel Phase Change Material Solar Dryer (PCMSD) using paraffin wax, stearic acid, and acetamide for drying tomatoes | Tomato | 2.51–2.98 years |
| Sharshir et al. [142] | Elsevier | 2024 | three types of solar air heaters (finned plate, evacuated tube, and modified evacuated tube solar air heaters) | Eggplant & Grapes | 1.27–8.41 years |
| Andharia et al. [143] | Elsevier | 2024 | A mixed-mode solar dryer equipped with a solar PV-powered dehumidifier | Agricultural products | 2.70 years |
| Zeeshan et al. [144] | ScienceDirect | 2024 | Indirectly forced convection desiccant integrated solar dryer | Tomato | 5.1396 years |
| Sharma et al. [145] | ScienceDirect | 2023 | Indirect type domestic hybrid solar dryer | Tomato | 4.21 years |
| Proposed dryer | ------ | | Forced air circulation, solar dryer integrated with solar tracking flat plate solar collector | Tomato | 2.40 years |

used to describe a sun-drying system's performance. To create and evaluate solar dryers and drying procedures, various modeling approaches have also been created. So, this article presents a thermo-environmental analysis and DK of a PV-operated tracking indirect solar dryer for tomato slices. The analysis of any thermal system has changed as a result of the application of modeling approaches in solar drying. Here, an effort is made to provide a comprehensive evaluation standard for solar dryers as well as a one-stop shop for users and researchers worldwide.

Based on obtained results, we found that the DT increased approximately by 1.667 to 1.6 times for the FCM and TCM systems, respectively, at constant air speed when the ST was changed from 4 to 8 mm. Where ST had a greater impact on DT than air speed.

Also, the MD were $7.15 \times 10^{-10}$ and $9.30 \times 10^{-10}$ m²/s for TCM and FCM systems, respectively at air speeds of 1.5 m/s and ST of 8 mm. as well as we concluded that, the Modified Two-Term II model was found to exhibit a good fit to the experimental data of different air speeds for drying TF at different ST using TCM and FCM systems. These findings were based on recorded observations. The models' $R^2$ values varied from 0.98005 to 0.99942 for the FCM system and from 0.99386 to 0.99976 for the TCM system.

On the other hand, regarding thermo-environmental analysis, we found that total EE of the solar dryer, collector and PV system was 1140.8 kW.h, maximum SEC was 8.15 and 5.42 kW.h/kg for FCM and TCM at 8.0 mm ST, annual thermal output of both solar dryer and PV system was calculated and showed that values was 375.34, 399.23, and 446.86 kW.h for the solar dryer integrated with FCM, and was 423.74, 422.88, and 474.85 kW.h for the solar dryer integrated with TCM at ST of 8, 6, and 4 mm, respectively. As well, the EPT was decreasing with decreasing the ST where the lowest EPT was observed at 4 mm ST. The $CO_2$ emission from the solar dryer was 137.11 tons while $CO_2$ mitigation per lifetime 5334.9, 5822.8 and 6795.4 tons for FCM and 6323.3, 6305.7 and 7366.9 tons for TCM. Minimum CCE of the solar dryer integrated with FCM and TCM was (26674, 29114 and 33977 \$), and (31616, 31528 and 36834 \$), respectively.

## 4.1. Future work

AI integration in solar drying technology holds immense potential for the future. By creating intelligent systems that adapt to changing conditions and optimize drying processes, AI can contribute to a more efficient, sustainable, and high-quality approach to solar drying.

## Author Contributions

**Conceptualization:** Daniel Eutyche Mbadjoun Wapet, Mostafa B. Mostafa, Khaled A. Metwally.

**Formal analysis:** Aml Abubakr Tantawy.

**Funding acquisition:** Daniel Eutyche Mbadjoun Wapet.

**Investigation:** Awad Ali Tayoush Oraiath, I. M. Elzein, Daniel Eutyche Mbadjoun Wapet, Aml Abubakr Tantawy.

**Methodology:** Abdallah Elshawadfy Elwakeel, Daniel Eutyche Mbadjoun Wapet, Khaled A. Metwally.

**Resources:** Awad Ali Tayoush Oraiath, Mohamed Metwally Mahmoud.

**Software:** Abdallah Elshawadfy Elwakeel, Daniel Eutyche Mbadjoun Wapet, Mostafa B. Mostafa.

**Supervision:** Mohsen A. Gameh, Ahmed S. Eissa.

**Validation:** Mahmoud M. Hussein.

**Writing – original draft:** Abdallah Elshawadfy Elwakeel, Daniel Eutyche Mbadjoun Wapet, Mostafa B. Mostafa, Khaled A. Metwally.

**Writing – review & editing:** Mohsen A. Gameh, Awad Ali Tayoush Oraiath, I. M. Elzein, Ahmed S. Eissa, Mohamed Metwally Mahmoud, Mahmoud M. Hussein.

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
