## [Decision Letter · Decision Letter 0]

1 May 2024

PONE-D-24-12484Drying Kinetics and Thermo-Environmental of a PV-Operated Tracking Indirect Solar Dryer for Tomato SlicesPLOS ONE

Dear Dr. Mbadjoun Wapet,

Thank you for submitting your manuscript to PLOS ONE. After careful consideration, we feel that it has merit but does not fully meet PLOS ONE’s publication criteria as it currently stands. Therefore, we invite you to submit a revised version of the manuscript that addresses the points raised during the review process.

In addition to reviewers' comments, the authors are suggested to mention accuracy of the devices/instruments. Also estimate errors in important quantities such as drying rate, moisture diffusivity and environmental parameters and include in revised paper .

We look forward to receiving your revised manuscript.

Kind regards,

Muhammad Shakaib, PhD

Academic Editor

PLOS ONE

Reviewers' comments:

Reviewer's Responses to Questions

**Comments to the Author**

1. Is the manuscript technically sound, and do the data support the conclusions?

Reviewer #1: Yes

Reviewer #2: Yes

2. Has the statistical analysis been performed appropriately and rigorously? 

Reviewer #1: Yes

Reviewer #2: Yes

3. Have the authors made all data underlying the findings in their manuscript fully available?

Reviewer #1: Yes

Reviewer #2: Yes

4. Is the manuscript presented in an intelligible fashion and written in standard English?

Reviewer #1: Yes

Reviewer #2: Yes

5. Review Comments to the Author

Reviewer #1: List of comments

The article “Drying Kinetics and Thermo-Environmental of a PV-Operated Tracking Indirect Solar Dryer for Tomato Slices” has been reviewed, and the following recommendations are made for its improvement.

1) Select proper keywords.,i.e replace tomato fruit with tomato slice

2) Drying time (DT) did not significantly change with increasing air speeds from 1 to 2 m/s.Justify it in the abstract.

3) In recent years, several researchers have used mathematical models for studying the DK of many agricultural products such as magical berries [45], figs [46], parboiled rice [47], pistachio nuts [48], olives [49], eggplant [50], tomato [51], [52], green peppers [53], strawberries [54], sweet potato and raisins [55], seedless grapes [56], pears [46], green peas [57], pistachio kernels [58], peach [59], orange peel [60], cassava [61], peach slices [62], mango [63], and grapes [64], where they were investigated and researched drying under thin layer (ThL) conditions

How your model is different from these models, justify it, and introduce a paragraph after ref.[64]

4) Expand section 2: Materials and methods

5) Expand section 1.1.4 by adding about “Non-linear regression analysis”

6) Explain figs.1 to 8 in a separate section.

7) Role of your model in reducing energy crisis and co2 emission.

8) Add some latest references in the introduction part for the years 2022-2023 including https://www.sciencedirect.com/science/article/abs/pii/S136403212300713X

9) Compare the Energy payback time (EPT) of your model with other similar models

10) Write the role of AI in solar dryers for future studies. This can be added in the conclusion for future scope

11) Overall presentation of the article is Good.

Reviewer #2: The manuscript proposed a PV-Operated Tracking Indirect Solar Dryer, and investigate the drying kinetics on tomato slices, it's interesting. However, some key factors should be addressed.

1.What's the innovation of the paper? tomato slices drying kinetices? A lot of published papers have address the kinetics, so what's new? The highlights or innovations should be given in the section of introduction.

2.If the the authors wants to do some contrast on the two dryer(tracking and no tracking), the performance should be investigated. But the author just give some drying kinetics, which is not related to dryer too much.

3.In the section of conclusion, please give more scientific evaluation.

6. PLOS authors have the option to publish the peer review history of their article (what does this mean?). If published, this will include your full peer review and any attached files.

Reviewer #1: No

Reviewer #2: No

---

## [Author Response · Author response to Decision Letter 0]

7 May 2024

***Technical response to the reviewers*** May 5th, 2024

Journal name: PLOS ONE 

No.: PONE-D-24-12484

OLD Title: “Drying Kinetics and Thermo-Environmental of a PV-Operated Tracking Indirect Solar Dryer for Tomato Slices”

NEW Title: “Drying Kinetics and Thermo-Environmental Analysis of a PV-Operated Tracking Indirect Solar Dryer for Tomato Slices” 

Abdallah E. Elwakeel1&*, Mohsen A. Gameh2, Ahmed S. Eissa3, I. M. Elzein4, Mohamed Metwally Mahmoud5, Daniel Eutyche Mbadjoun Wapet6&*, Mahmoud M. Hussein5,7, Hany S. Hussein8,9, Aml Abubakr Tantawy10, Mostafa B. Mostafa1, Khaled A. Metwally11

1 Agricultural Engineering Department, Faculty of Agriculture and Natural Resources, Aswan University, Aswan 81528, Egypt, Abdallah_elshawadfy@agr.aswu.edu.eg; Mostafabadawy017@gmail.com

2 Soils and Water Department, Faculty of Agriculture, Assiut University, Assiut 71526, Egypt; ali28@aun.edu.eg

3Agricultural Products Process Engineering Department, Faculty of Agricultural Engineering, Al-Azhar University, Cairo 11751, Egypt; Ahmedeissa2205.el@azhar.edu.eg

4Department of Electrical engineering, University of Doha for Science and Technology, Doha, Qatar

60101973@udst.edu.qa

5Electrical Engineering Department, Faculty of Energy Engineering, Aswan University, Aswan 81528, Egypt, Metwally_M@aswu.edu.eg

6&* National Advanced School of Engineering, Universit´e de Yaound´e I, Yaound´e, Cameroon, eutychedan@gmail.com

7Department of Communications Technology Engineering, Technical College, Imam Ja’afar Al-Sadiq University, Baghdad, 10053, Iraq, mahmoud_hussein@aswu.edu.eg

8Electrical Engineering Department, College of Engineering, King Khalid University, Abha 62529, Saudi Arabia

9Electrical Power Engineering Department, Faculty of Engineering, Aswan University, Aswan 81528, Egypt,

hany.hussein@aswu.edu.eg

10Food Science and Technology Department, Faculty of Agriculture and Natural Resources, Aswan University, Aswan 81528, Egypt, Aml.abubakr@agr.aswu.edu.eg

11Soil and Water Sciences Department, Faculty of Technology and Development, Zagazig University,

Zagazig 44511, Egypt; khametwally@zu.edu.eg

*Corresponding author: Daniel Eutyche Mbadjoun Wapet, eutychedan@gmail.com

Dear Editors and Reviewers

The authors are thankful to the learned Editors and Reviewers for their thoughtful and detailed comments to improve the quality of the manuscript. The authors have tried to address all the concerns, and the corrections are incorporated in the revised manuscript. The replies to the reviewer’s comments are provided below.

We hope that this revised version can meet the reviewer’s expectations and the standards for publication in PLOS ONE Journal.

The changes incorporated in the revised manuscript are highlighted in YELLOW.

Editor's Comments:

Our sincere thanks and appreciation to the Editors for recommending the submission of the revised manuscript with major revision. To improve the quality of the manuscript, the reviewer's queries are addressed, and their suggestions are incorporated into the revised manuscript. 

Reviewer Comments:

Reviewer 1:

Comments to the Authors:

Comments1: Select proper keywords.,i.e replace tomato fruit with tomato slice

Response1: The authors are extremely thankful to the reviewer for this thoughtful point. We agree with you, and we corrected it, kindly check the updated paper (keywords). 

Comment 2: Drying time (DT) did not significantly change with increasing air speeds from 1 to 2 m/s. Justify it in the abstract.

Response 2: The authors are extremely thankful to the reviewer for this thoughtful point. We agree with you, and we added it, kindly check the updated paper (Abstract).

Comment 3: In recent years, several researchers have used mathematical models for studying the DK of many agricultural products such as magical berries [45], figs [46], parboiled rice [47], pistachio nuts [48], olives [49], eggplant [50], tomato [51], [52], green peppers [53], strawberries [54], sweet potato and raisins [55], seedless grapes [56], pears [46], green peas [57], pistachio kernels [58], peach [59], orange peel [60], cassava [61], peach slices [62], mango [63], and grapes [64], where they were investigated and researched drying under thin layer (ThL) conditions. How your model is different from these models, justify it, and introduce a paragraph after ref. [64]

Response 3: The authors are extremely thankful to the reviewer for this thoughtful point. We agree with you, and we added it, kindly check the updated paper (after Ref. 74).

Comment 4: Expand section 2: Materials and methods

Response 4: The authors are extremely thankful to the reviewer for this thoughtful point. The authors completely agree with you, but as explained in subtitle 2.2. in the updated paper, this study is complementing a previous study by Elwakeel et al. [16]

Ref. Elwakeel, A. E., Mostafa B. Mostafa, Mohsen A. G., and Ahmed S. E. "Some Engineering Factors Affecting Utilization of Solar Energy in Drying Tomato Fruits." Aswan University Journal of Environmental Studies 5, no. 1 (2024): 52-68.

Comment 5: Expand section 1.1.4 by adding about “Non-linear regression analysis”.

Response 5: The authors are extremely thankful to the reviewer for this thoughtful point. The authors completely agree with you, and we added it, kindly check the updated paper (page 7).

Comment 6: Explain figs.1 to 8 in a separate section.

Response 6: The authors are extremely thankful to the reviewer for this thoughtful point. We completely agree with you, but we have written the paper depends on the PLOS ONE journal format. 

Comment 7: Role of your model in reducing energy crisis and CO2 emission.

Response 7: The authors are extremely thankful to the reviewer for this thoughtful point. The role of the solar dryer for reducing or mitigation energy and CO2 per lifetime. The annual thermal output of both solar dryer and PV system was calculated and showed that values was 375.34, 399.23, and 446.86 kW.h for the solar dryer integrated with FCM, and was 423.74, 422.88, and 474.85 kW.h for the solar dryer integrated with TCM at ST of 8, 6, and 4 mm, respectively. As well, CO2 per lifetime was 5334.9, 5822.8 and 6795.4 tons for FCM and 6323.3, 6305.7 and 7366.9 tons for TCM, at tomato slice thicknesses of 8, 6, and 4 mm, respectively. Kindly check the updated paper (Table 5 & page 20).

Comment 8: Add some latest references in the introduction part for the years 2022-2023 including https://www.sciencedirect.com/science/article/abs/pii/S136403212300713X

Response 8: The authors are extremely thankful to the reviewer for this thoughtful point. We completely agree with you, and we added many recent paper most of them in the years 2023 & 2024, kindly check the updated paper.

Comment 9: Compare the Energy payback time (EPT) of your model with other similar models.

Response 9: The authors are extremely thankful to the reviewer for this thoughtful point. We completely agree with you, kindly check the updated paper (Table 6).

Comment 10: rite the role of AI in solar dryers for future studies. This can be added in the conclusion for future scope.

Response 10: The authors are extremely thankful to the reviewer for this thoughtful point. We completely agree with you, and we added the role of AI in solar dryers for future studies. Kindly check the updated paper (Conclusion and future work).

Comment 11: Overall presentation of the article is Good.

Response 11: The authors are thankful to the honorable reviewer for the words of encouragement and trust in our work. 

Reviewer 2:

The authors are thankful to the honorable reviewer for the words of encouragement and trust in our work.

Comment-1: What's the innovation of the paper? tomato slices drying kinetices? A lot of published papers have address the kinetics, so what's new? The highlights or innovations should be given in the section of introduction.

Response-1: The authors are extremely thankful to the reviewer for this thoughtful point. We completely agree with you, and we added it to the introduction section, kindly check the updated paper.

Comment-2: If the the authors wants to do some contrast on the two dryer (tracking and no tracking), the performance should be investigated. But the author just give some drying kinetics, which is not related to dryer too much.

Response-2: The authors are extremely thankful to the reviewer for this thoughtful point. The authors completely agree with you, but as explained in subtitle 2.2. in the updated paper, this study is complementing a previous study by Elwakeel et al. [16]

Ref. Elwakeel, A. E., Mostafa B. Mostafa, Mohsen A. G., and Ahmed S. E. "Some Engineering Factors Affecting Utilization of Solar Energy in Drying Tomato Fruits." Aswan University Journal of Environmental Studies 5, no. 1 (2024): 52-68.

While the current study was undertaken to discern the drying performance by utilizing both solar dryers integrated with FCM and TCM systems. By fitting drying curves with well-known models, the best model can be determined. Furthermore, calculate mathematical modeling. Finally, an environmental and energy analysis was performed to determine the energy payback time, and net CO2 mitigation over the lifetime of the developed solar dryer. Through this study, we hope to add to the body of knowledge already available on the drying kinetics of tomato fruit varieties and shed light on whether using solar dryers for this purpose would be environmentally feasible.

Comment-3: In the section of conclusion, please give more scientific evaluation.

Response-3: The authors are extremely thankful to the reviewer for this thoughtful point. We agree with you, so we developed the conclusion section. Kindly check the updated paper (conclusion section).

The authors once again thank the learned Editors and Reviewers for their valuable comments for improving the quality of the manuscript.

---

## [Decision Letter · Decision Letter 1]

21 May 2024

PONE-D-24-12484R1Drying Kinetics and Thermo-Environmental Analysis of a PV-Operated Tracking Indirect Solar Dryer for Tomato SlicesPLOS ONE

Dear Dr. Mbadjoun Wapet,

Thank you for submitting your manuscript to PLOS ONE. After careful consideration, we feel that it has merit but does not fully meet PLOS ONE’s publication criteria as it currently stands. Therefore, we invite you to submit a revised version of the manuscript that addresses the points raised during the review process.

The authors are suggested to include comparison of the previous studies (including their own ones) and highlight the new methods and findings of the present work.

As in the previous comments, also mention accuracy of the devices/instruments. Further, estimate errors in important quantities such as drying rate, moisture diffusivity and environmental parameters

We look forward to receiving your revised manuscript.

Kind regards,

Muhammad Shakaib, PhD

Academic Editor

PLOS ONE

Reviewers' comments:

Reviewer's Responses to Questions

**Comments to the Author**

1. If the authors have adequately addressed your comments raised in a previous round of review and you feel that this manuscript is now acceptable for publication, you may indicate that here to bypass the “Comments to the Author” section, enter your conflict of interest statement in the “Confidential to Editor” section, and submit your "Accept" recommendation.

Reviewer #1: All comments have been addressed

Reviewer #2: (No Response)

2. Is the manuscript technically sound, and do the data support the conclusions?

Reviewer #1: Yes

Reviewer #2: (No Response)

3. Has the statistical analysis been performed appropriately and rigorously? 

Reviewer #1: Yes

Reviewer #2: (No Response)

4. Have the authors made all data underlying the findings in their manuscript fully available?

Reviewer #1: Yes

Reviewer #2: (No Response)

5. Is the manuscript presented in an intelligible fashion and written in standard English?

Reviewer #1: Yes

Reviewer #2: (No Response)

6. Review Comments to the Author

Reviewer #1: “The authors have addressed all the comments in the manuscript PONE-D-24-12484R1, entitled ‘Drying Kinetics and Thermo-Environmental Analysis of a PV-Operated Tracking Indirect Solar Dryer for Tomato Slices.’ The paper is accepted in its present form.”

Reviewer #2: The author explains many times that some works have been done in a previous study by Elwakeel et al. I am not agree with that. How much innovation are there in the paper? And what is the difference with the previous study?

7. PLOS authors have the option to publish the peer review history of their article (what does this mean?). If published, this will include your full peer review and any attached files.

Reviewer #1: No

Reviewer #2: No

---

## [Author Response · Author response to Decision Letter 1]

23 May 2024

***Technical response to the reviewers*** May 23th, 2024

Journal name: PLOS ONE 

No.: PONE-D-24-12484

OLD Title: “Drying Kinetics and Thermo-Environmental of a PV-Operated Tracking Indirect Solar Dryer for Tomato Slices”

NEW Title: “Drying Kinetics and Thermo-Environmental Analysis of a PV-Operated Tracking Indirect Solar Dryer for Tomato Slices” 

Abdallah E. Elwakeel1&*, Mohsen A. Gameh2, Ahmed S. Eissa3, I. M. Elzein4, Mohamed Metwally Mahmoud5, Daniel Eutyche Mbadjoun Wapet6&*, Mahmoud M. Hussein5,7, Hany S. Hussein8,9, Aml Abubakr Tantawy10, Mostafa B. Mostafa1, Khaled A. Metwally11

1 Agricultural Engineering Department, Faculty of Agriculture and Natural Resources, Aswan University, Aswan 81528, Egypt, Abdallah_elshawadfy@agr.aswu.edu.eg; Mostafabadawy017@gmail.com

2 Soils and Water Department, Faculty of Agriculture, Assiut University, Assiut 71526, Egypt; ali28@aun.edu.eg

3Agricultural Products Process Engineering Department, Faculty of Agricultural Engineering, Al-Azhar University, Cairo 11751, Egypt; Ahmedeissa2205.el@azhar.edu.eg

4Department of Electrical engineering, University of Doha for Science and Technology, Doha, Qatar

60101973@udst.edu.qa

5Electrical Engineering Department, Faculty of Energy Engineering, Aswan University, Aswan 81528, Egypt, Metwally_M@aswu.edu.eg

6&* National Advanced School of Engineering, Universit´e de Yaound´e I, Yaound´e, Cameroon, eutychedan@gmail.com

7Department of Communications Technology Engineering, Technical College, Imam Ja’afar Al-Sadiq University, Baghdad, 10053, Iraq, mahmoud_hussein@aswu.edu.eg

8Electrical Engineering Department, College of Engineering, King Khalid University, Abha 62529, Saudi Arabia

9Electrical Power Engineering Department, Faculty of Engineering, Aswan University, Aswan 81528, Egypt,

hany.hussein@aswu.edu.eg

10Food Science and Technology Department, Faculty of Agriculture and Natural Resources, Aswan University, Aswan 81528, Egypt, Aml.abubakr@agr.aswu.edu.eg

11Soil and Water Sciences Department, Faculty of Technology and Development, Zagazig University,

Zagazig 44511, Egypt; khametwally@zu.edu.eg

*Corresponding author: Daniel Eutyche Mbadjoun Wapet, eutychedan@gmail.com

Dear Editors and Reviewers

The authors are thankful to the learned Editors and Reviewers for their thoughtful and detailed comments to improve the quality of the manuscript. The authors have tried to address all the concerns, and the corrections are incorporated in the revised manuscript. The replies to the reviewer’s comments are provided below.

We hope that this revised version can meet the reviewer’s expectations and the standards for publication in PLOS ONE Journal.

The changes incorporated in the revised manuscript are highlighted in YELLOW.

Editor's Comments:

Our sincere thanks and appreciation to the Editors for recommending the submission of the revised manuscript with major revision. To improve the quality of the manuscript, the reviewer's queries are addressed, and their suggestions are incorporated into the revised manuscript. 

Comment-1: The authors are suggested to include comparison of the previous studies (including their own ones) and highlight the new methods and findings of the present work.

Response-1: The authors are extremely thankful to the reviewer for this thoughtful point. We want to clarify that, the current study is part of a project that aims to improve the use of solar energy in drying tomato fruits. It involves comparing a solar dryer equipped with a solar tracking unit and a solar dryer equipped with a fixed solar collector. Earlier this year, only one research from this project was published, containing engineering details related to the designs of both systems and a comparison of their performance including (1. Components; 2. Design; 3. measurements of ambient air, temperature of ambient air and hot air inside the drying room; 4. Effect of hot air velocity on the reduction on the total weight of the dried tomato fruit samples; 5. Effect of thicknesses of TS on the reduction on the total weight of the dried TF samples; 6. Thermal efficiency of the solar collector; 7. Thermal efficiency of the PV system), [Ref. no. 16: Eissa AS, Gameh MA, Mostafa MB, Elwakeel AE. Some Engineering Factors Affecting Utilization of Solar Energy in Drying Tomato Fruits Introduction. 2024;5: 52–68. doi:10.21608/aujes.2024.252750.1202]. The purpose of the current article is to deepen the comparative study between both systems, as described in the introduction [By fitting drying curves with well-known models, the best model can be determined. Furthermore, calculate mathematical modeling. Finally, an environmental and energy analysis was performed to determine the energy payback time, and net CO2 mitigation over the lifetime of the developed solar dryer. Through this study, we hope to add to the body of knowledge already available on the drying kinetics of tomato fruit varieties and shed light on whether using solar dryers for this purpose would be environmentally feasible]. The previous reference was mentioned in the materials and methods section to provide background on the previous research, which is an integral part of the project, and to avoid prolonging the explanation of the engineering points and technological evaluation of both systems in the current research. Focus only on achieving the objectives of the current research.

Accordingly, there is only one published research related to the current work, which is completely different from the current work, and there are no common points between the two researches that can be discussed and compared.

Comment-2: As in the previous comments, also mention accuracy of the devices/instruments. Further, estimate errors in important quantities such as drying rate, moisture diffusivity and environmental parameters.

Response-2: The authors are extremely thankful to the editor for this thoughtful point. Where we added the comparison through table 2.

Reviewer Comments:

Reviewer 2:

Comment-1: The author explains many times that some works have been done in a previous study by Elwakeel et al. I am not agree with that. How much innovation are there in the paper? And what is the difference with the previous study?

Response-1: The authors are extremely thankful to the reviewer for this thoughtful point. We want to clarify that, the current study is part of a project that aims to improve the use of solar energy in drying tomato fruits. It involves comparing a solar dryer equipped with a solar tracking unit and a solar dryer equipped with a fixed solar collector. Earlier this year, only one research from this project was published, containing engineering details related to the designs of both systems and a comparison of their performance including (1. Components; 2. Design; 3. measurements of ambient air, temperature of ambient air and hot air inside the drying room; 4. Effect of hot air velocity on the reduction on the total weight of the dried tomato fruit samples; 5. Effect of thicknesses of TS on the reduction on the total weight of the dried TF samples; 6. Thermal efficiency of the solar collector; 7. Thermal efficiency of the PV system), [Ref. no. 16: Eissa AS, Gameh MA, Mostafa MB, Elwakeel AE. Some Engineering Factors Affecting Utilization of Solar Energy in Drying Tomato Fruits Introduction. 2024;5: 52–68. doi:10.21608/aujes.2024.252750.1202]. The purpose of the current article is to deepen the comparative study between both systems, as described in the introduction [By fitting drying curves with well-known models, the best model can be determined. Furthermore, calculate mathematical modeling. Finally, an environmental and energy analysis was performed to determine the energy payback time, and net CO2 mitigation over the lifetime of the developed solar dryer. Through this study, we hope to add to the body of knowledge already available on the drying kinetics of tomato fruit varieties and shed light on whether using solar dryers for this purpose would be environmentally feasible]. The previous reference was mentioned in the materials and methods section to provide background on the previous research, which is an integral part of the project, and to avoid prolonging the explanation of the engineering points and technological evaluation of both systems in the current research. Focus only on achieving the objectives of the current research.

Accordingly, there is only one published research related to the current work, which is completely different from the current work, and there are no common points between the two researches that can be discussed and compared.

The authors once again thank the learned Editors and Reviewers for their valuable comments for improving the quality of the manuscript.

---

## [Decision Letter · Decision Letter 2]

5 Jun 2024

PONE-D-24-12484R2Drying Kinetics and Thermo-Environmental Analysis of a PV-Operated Tracking Indirect Solar Dryer for Tomato SlicesPLOS ONE

Dear Dr. Mbadjoun Wapet,

Thank you for submitting your manuscript to PLOS ONE. After careful consideration, we feel that it has merit but does not fully meet PLOS ONE’s publication criteria as it currently stands. Therefore, we invite you to submit a revised version of the manuscript that addresses the points raised during the review process.

The authors are recommended to determine errors in the important quantities such as MR and EE and include in the revised paper. Further, the results should be shown either in Table or Figure (not both) as also as suggested by the reviewer.

We look forward to receiving your revised manuscript.

Kind regards,

Muhammad Shakaib, PhD

Academic Editor

PLOS ONE

Journal Requirements:

Reviewers' comments:

Reviewer's Responses to Questions

**Comments to the Author**

1. If the authors have adequately addressed your comments raised in a previous round of review and you feel that this manuscript is now acceptable for publication, you may indicate that here to bypass the “Comments to the Author” section, enter your conflict of interest statement in the “Confidential to Editor” section, and submit your "Accept" recommendation.

Reviewer #2: All comments have been addressed

2. Is the manuscript technically sound, and do the data support the conclusions?

Reviewer #2: (No Response)

3. Has the statistical analysis been performed appropriately and rigorously? 

Reviewer #2: No

4. Have the authors made all data underlying the findings in their manuscript fully available?

Reviewer #2: (No Response)

5. Is the manuscript presented in an intelligible fashion and written in standard English?

Reviewer #2: (No Response)

6. Review Comments to the Author

Reviewer #2: Although the authors have revised the manuscript, which have not address the points.

1. In the section of 2.6 data measuring, it's not so important on the error. According to the error propagation theory, what's the error of MR, and EE?

2. The results in Table 4 are the same with the Fig, 6 and 7? if so, I think you can choose one kind of them.

3. Especailly, what's the advantage using the kind of dryer? save energy? what should be highlighten.

7. PLOS authors have the option to publish the peer review history of their article (what does this mean?). If published, this will include your full peer review and any attached files.

Reviewer #2: No

---

## [Author Response · Author response to Decision Letter 2]

10 Jun 2024

***Technical response to the reviewers*** Jun 11th, 2024

Journal name: PLOS ONE 

No.: PONE-D-24-12484

Drying Kinetics and Thermo-Environmental Analysis of a PV-Operated Tracking Indirect Solar Dryer for Tomato Slices

Abdallah Elshawadfy Elwakeel1&*, Mohsen A. Gameh2, Awad Ali Tayoush Oraiath3, I. M. Elzein4, Ahmed S. Eissa5, Mohamed Metwally Mahmoud6, Daniel Eutyche Mbadjoun Wapet7&*, Mahmoud M. Hussein6,8, Aml Abubakr Tantawy9, Mostafa B. Mostafa1, Khaled A. Metwally10

1 Agricultural Engineering Department, Faculty of Agriculture and Natural Resources, Aswan University, Aswan 81528, Egypt, Abdallah_elshawadfy@agr.aswu.edu.eg; Mostafabadawy017@gmail.com

2 Soils and Water Department, Faculty of Agriculture, Assiut University, Assiut 71526, Egypt; ali28@aun.edu.eg

3 Department of Agricultural Engineering, Faculty of Agriculture, Omar Al Mukhtar University, Libya 991, awad.ali@omu.edu.ly

4Department of Electrical engineering, University of Doha for Science and Technology, Doha, Qatar

60101973@udst.edu.qa

5Agricultural Products Process Engineering Department, Faculty of Agricultural Engineering, Al-Azhar University, Cairo 11751, Egypt; Ahmedeissa2205.el@azhar.edu.eg

6Electrical Engineering Department, Faculty of Energy Engineering, Aswan University, Aswan 81528, Egypt, Metwally_M@aswu.edu.eg

7&* National Advanced School of Engineering, Universit´e de Yaound´e I, Yaound´e, Cameroon, eutychedan@gmail.com

8 Department of Communications Technology Engineering, Technical College, Imam Ja’afar Al-Sadiq University, Baghdad, 10053, Iraq, mahmoud_hussein@aswu.edu.eg

9 Food Science and Technology Department, Faculty of Agriculture and Natural Resources, Aswan University, Aswan 81528, Egypt, Aml.abubakr@agr.aswu.edu.eg

10 Soil and Water Sciences Department, Faculty of Technology and Development, Zagazig University,

Zagazig 44519, Egypt; khametwally@zu.edu.eg

*Corresponding author: Daniel Eutyche Mbadjoun Wapet, eutychedan@gmail.com

Dear Editors and Reviewers

The authors are thankful to the learned Editors and Reviewers for their thoughtful and detailed comments to improve the quality of the manuscript. The authors have tried to address all the concerns, and the corrections are incorporated in the revised manuscript. The replies to the reviewer’s comments are provided below.

We hope that this revised version can meet the reviewer’s expectations and the standards for publication in PLOS ONE Journal.

The changes incorporated in the revised manuscript are highlighted in YELLOW.

Editor's Comments:

Our sincere thanks and appreciation to the Editors for recommending the submission of the revised manuscript with major revision. To improve the quality of the manuscript, the reviewer's queries are addressed, and their suggestions are incorporated into the revised manuscript. 

Reviewer Comments:

Reviewer 2:

Comment-1: In the section of 2.6 data measuring, it's not so important on the error. According to the error propagation theory, what's the error of MR, and EE?

Response-1: The authors are extremely thankful to the reviewer for this thoughtful point. The error of MR, and EE were added to the illustrated data in Table 2. Kindly check the developed manuscript.

Comment-2: The results in Table 4 are the same with the Fig, 6 and 7? if so, I think you can choose one kind of them.

Response-2: The authors are extremely thankful to the reviewer for this thoughtful point. Where we deleted these figures. Kindly check the developed manuscript.

Comment-3: Especailly, what's the advantage using the kind of dryer? save energy? what should be highlighten.

Response-3: The authors are extremely thankful to the reviewer for this thoughtful point. 

The solar dryer with a solar tracking collector offers significant advantages in terms of energy collected compared to the other solar dryer with a fixed collector:

• Increased Solar Exposure: The primary benefit is that a tracking collector can continuously orient itself towards the sun throughout the day. This maximizes the amount of solar radiation hitting the collector, leading to a significant increase in the heat energy collected. Fixed collectors, on the other hand, can only capture optimal sunlight for a limited period, typically around midday.

• Improved Drying Efficiency: With more collected heat energy, a tracking collector solar dryer can achieve faster drying times and higher drying capacity. This translates to improved overall drying efficiency.

• Reduced Drying Time: Faster drying times minimize the risk of spoilage for products being dried, especially food items. This is crucial for preserving the quality of the dried goods.

• Potential for year-round operation: Depending on the location and climate, a tracking collector can potentially extend the operational season of the solar dryer. By capturing more sunlight even during off-peak sun hours, tracking collectors can be more effective in areas with shorter daylight periods.

The authors once again thank the learned Editors and Reviewers for their valuable comments for improving the quality of the manuscript.

---

## [Editor Report · Decision Letter 3]

15 Jun 2024

Drying Kinetics and Thermo-Environmental Analysis of a PV-Operated Tracking Indirect Solar Dryer for Tomato Slices

PONE-D-24-12484R3

Dear Dr. Mbadjoun Wapet,

We’re pleased to inform you that your manuscript has been judged scientifically suitable for publication and will be formally accepted for publication once it meets all outstanding technical requirements.

Kind regards,

Muhammad Shakaib, PhD

Academic Editor

PLOS ONE
---

## [Editor Report · Acceptance letter]

9 Jul 2024

PONE-D-24-12484R3 

PLOS ONE

Dear Dr. Mbadjoun Wapet, 

I'm pleased to inform you that your manuscript has been deemed suitable for publication in PLOS ONE. Congratulations! Your manuscript is now being handed over to our production team.

Kind regards, 

on behalf of

Dr. Muhammad Shakaib 

Academic Editor

PLOS ONE